**Investigation**

# Characterization of the *Pristionchus pacificus* "epigenetic toolkit" reveals the evolutionary loss of the histone methyltransferase complex PRC2

Audrey L. Brown [ID],[1] Adriaan B. Meiborg [ID],[2,3] Mirita Franz-Wachtel,[4] Boris Macek,[4] Spencer Gordon,[1] Ofer Rog [ID],[1] Cameron J. Weadick,[5] Michael S. Werner [ID] [1,*]

[1]School of Biological Sciences, The University of Utah, Salt Lake City, UT 84112, USA
[2]Developmental Biology Unit, European Molecular Biology Laboratory (EMBL), 69117 Heidelberg, Germany
[3]Faculty of Biosciences, Collaboration for joint PhD degree between EMBL and Heidelberg University, 69120 Heidelberg, Germany
[4]Proteome Center Tübingen, University of Tübingen, 72074 Tübingen, Germany
[5]Biosciences, University of Exeter, Exeter EX4 4QD, UK

*Corresponding author: School of Biological Sciences, University of Utah, 257 S 1400 E, Salt Lake City, UT 84112, USA. Email: michael.werner@utah.edu

Comparative approaches have revealed both divergent and convergent paths to achieving shared developmental outcomes. Thus, only through assembling multiple case studies can we understand biological principles. Yet, despite appreciating the conservation—or lack thereof—of developmental networks, the conservation of epigenetic mechanisms regulating these networks is poorly understood. The nematode *Pristionchus pacificus* has emerged as a model system of plasticity and epigenetic regulation as it exhibits a bacterivorous or omnivorous morph depending on its environment. Here, we determined the "epigenetic toolkit" available to *P. pacificus* as a resource for future functional work on plasticity, and as a comparison with *Caenorhabditis elegans* to investigate the conservation of epigenetic mechanisms. Broadly, we observed a similar cast of genes with putative epigenetic function between *C. elegans* and *P. pacificus*. However, we also found striking differences. Most notably, the histone methyltransferase complex PRC2 appears to be missing in *P. pacificus*. We described the deletion/pseudogenization of the PRC2 genes *mes-2* and *mes-6* and concluded that both were lost in the last common ancestor of *P. pacificus* and a related species *P. arcanus*. Interestingly, we observed the enzymatic product of PRC2 (H3K27me3) by mass spectrometry and immunofluorescence, suggesting that a currently unknown methyltransferase has been co-opted for heterochromatin silencing. Altogether, we have provided an inventory of epigenetic genes in *P. pacificus* to compare with *C. elegans*. This inventory will enable reverse-genetic experiments related to plasticity and has revealed the first loss of PRC2 in a multicellular organism.

Keywords: epigenetics; *Pristionchus pacificus*; *Caenorhabditis elegans*; gene loss; evo-devo; PRC2; comparative analysis; bioinformatics; chromatin; histones

## Introduction

Comparative studies have revealed how developmental networks can evolve over time; sometimes generating the same phenotype, and sometimes leading to novel traits (Peel 2008; Peter and Davidson 2011). These studies have primarily focused on the conservation of genes underlying fixed trait development. Meanwhile, comparatively little attention has been paid to the evolution of factors that regulate environmentally sensitive development. Though the term "epigenetic" has taken on various meanings over time, the current definition refers to mitotically and/or meiotically stable, non-DNA sequence based mechanisms that regulate gene expression (Jablonka and Lamb 2002; Deans and Maggert 2015). Three distinct mechanisms fall under this definition: DNA modification (e.g. 5mC), histone posttranslational modification (e.g. H3K27me3), and noncoding RNA pathways (e.g. RNAi) (Liebers *et al.* 2014; Allis and Jenuwein 2016). Studying the evolution of epigenetic pathways may offer insights into developmental robustness vs plasticity and the evolution of novel traits (Levis and Pfennig 2019).

The nematode *Pristionchus pacificus* was established in the 1990s as a comparative system to the model nematode *Caenorhabditis elegans* (last sharing a common ancestor 80–200 mya) (Howard *et al.* 2022). Having two divergent, experimentally tractable model organisms in the same order has facilitated functional evolutionary and developmental (evo-devo) studies leading to several unexpected findings (Markov *et al.* 2016; Rošić *et al.* 2018; Beltran *et al.* 2019; Bui and Ragsdale 2019; Hong *et al.* 2019; Weadick 2020; Sun *et al.* 2021; Lo *et al.* 2022). For example, early studies on vulva development revealed the extent of developmental systems drift, whereby the genetic underpinnings of a trait diverge even as the trait itself is conserved (Wang and Sommer 2011; Sommer 2012). However, a thorough exploration of the conservation of epigenetic pathways between the two species has not yet been done.

There is also interest in characterizing the epigenetic pathways in *P. pacificus* through the lens of developmental plasticity, a phenomenon where environmental conditions experienced during development influence adult phenotypes. *P. pacificus* displays morphological plasticity of its feeding structures: adult worms

exhibit either an omnivorous or bacterivorous mouth-form depending on signals experienced as juveniles (Bento *et al.* 2010). The difference between these forms is multifaceted: the microbivorous morph is narrow, deep, and contains a single dorsal "tooth," while the omnivorous morph is wide, shallow, and contains two teeth. Several studies have now pointed to epigenetic mechanisms as a way to translate environmental signals into gene expression changes that underly the development of different traits (Valena and Moczek 2012; Duncan *et al.* 2014; Yan *et al.* 2014; Kilvitis *et al.* 2017; Thorson *et al.* 2017; Budd *et al.* 2022; Toker *et al.* 2022; Jordan *et al.* 2023; Werner *et al.* 2023). We recently showed that perturbing histone acetylation/deacetylation with chemical inhibitors alters both mouth-form phenotype and the expression of key "switch genes," indicating that epigenetic pathways are, at least in part, responsible for the regulation of mouth-form development (Werner *et al.* 2023). Additionally, the histone demethylase *spr-5/KDM1A/LSD1* was found to regulate environmental sensitivity and morphology of mouth forms (Levis and Ragsdale 2023). However, further genetic and biochemical experiments are needed to reveal how histone modifications regulate plasticity in this system, which requires prior knowledge of the "writers" and "erasers" of modifications present in *P. pacificus*. In essence, in order to manipulate proteins that effect epigenetic modifications, we first need to know which those proteins and modifications are.

As a relatively new model system, *P. pacificus* lacks a well-characterized "toolkit" of genes and modifications with epigenetic function. A high-quality chromosome-scale genome assembly with manually curated gene annotations has been created for *P. pacificus* (Rödelsperger *et al.* 2017; Athanasouli *et al.* 2020), yet the annotations lack thorough functional categorization compared to other model systems backed by decades of experimental research. To address this gap, we identified putative *P. pacificus* epigenetic genes using an orthology and domain-informed annotation pipeline. We then used LC-MS/MS to identify the histone posttranslational modifications (PTMs) present in *P. pacificus* and, by orthology, predict which proteins may have added or removed these observed marks. Surprisingly, after curating this inventory, we found that the highly conserved methyltransferase PRC2 (Polycomb Repressive Complex 2) was missing in *P. pacificus* despite the presence of its associated mark, H3K27me3. We investigated the presence/absence of the PRC2 complex in the *Pristionchus* phylogeny and traced the loss of two of its components (MES-2 and MES-6) to the last common ancestor of *P. pacificus* and its relative *P. arcanus*. To our knowledge, its absence in *P. pacificus* represents the first identified loss in a multicellular organism. The presence of H3K27me3 in the absence of PRC2 indicates that a currently unknown methyltransferase is responsible for H3K27me3 and maintains its role in gene silencing. In summary, we produced a dataset of the "epigenetic toolkit" available to *P. pacificus* by identifying enzymes that "write" and "erase" epigenetic modifications, remodel chromatin, or are involved in small RNA biogenesis. We envision this inventory as a resource for future experimental work on developmental plasticity and comparative studies with *C. elegans*—including the evolution of H3K27 methyltransferases.

## Methods
### Identification of putative epigenetic genes

We used a domain and orthology-informed pipeline adapted from Pratx *et al.* 2018 to predict genes in *P. pacificus* and *C. elegans* with putative epigenetic function. As input, we used the proteomes of 17 species, including six well-studied model organisms (*C. elegans*, *Drosophila melanogaster*, *Homo sapiens*, *Arabidopsis thaliana*, *Saccharomyces cerevisiae*, *Schyzosaccharomyces pombe*) (Davis *et al.* 2022; Bult and Sternberg 2023), two emerging model systems for developmental plasticity (*Acyrthosiphon pisum*, *Apis mellifera*), six additional nematodes (*P. pacificus*, *Pristionchus exspectatus*, *Pristionchus mayeri*, *Brugia malayi*, *Strongyloides ratti*, *Trichinella spiralis*), plus three additional species—an animal (*Danio rerio*), fungus (*Leptosphaeria maculans*), and plant (*Selaginella moellendorffii*)—included to help improve phylogenetic resolution. Proteome sources and accession numbers are indicated in Supplementary Table 1 (Athanasouli *et al.* 2020; Rödelsperger 2021; UniProt Consortium 2021).

To identify putative epigenetic genes in our target species (*P. pacificus*), we relied on reference datasets of known epigenetic proteins and epigenetic-associated protein domains (i.e. a protein or domain previously shown to be exclusively linked to epigenetic processes) in the six model species included in our above input (*C. elegans*, *H. sapiens*, *A. thaliana*, *S. cerevisiae*, *S. pombe*, and *D. melanogaster*). We initially obtained these reference datasets from Pratx *et al.* (2018). However, we adjusted them to reflect updated UniProt accession numbers and to include an additional 83 epigenetic proteins that we identified from UniProt and additional literature and database searches on top of the original 691 from Pratx *et al.* 2018 (Supplementary Files 1 and 2).

The pipeline consists of six steps (Fig. 1):

1) A Pfam domain search on all *P. pacificus* proteins using InterProScan software, version 5.58-91.0, with default search settings (Supplementary File 3; Jones *et al.* 2014).
2) Isolation of *P. pacificus* proteins containing epigenetic-associated Pfam domains (based on the reference epigenetic domain dataset; Supplementary File 1).
3) Orthology clustering of the 17 proteomes using OrthoFinder version 2.5.4 software, default settings (Emms and Kelly 2019). From OrthoFinder, we obtained "orthogroups" (i.e. groups of orthologous proteins) and orthogroup gene phylogenies (Supplementary File 4).
4) Isolation of *P. pacificus* proteins from orthogroups containing one or more known epigenetic proteins (based on the reference epigenetic-protein dataset; Supplementary File 4).
5) Quality control to remove false positives, including validating orthogroup phylogenies and orthogroup domain composition, and reannotation of gene annotation errors (Supplementary File 5). Additional details on quality control steps are provided below.
6) Merging the orthology-identified and domain-identified *P. pacificus* protein datasets and assembly of a final epigenetic gene dataset from the genes encoding each protein. This was done to account for cases where multiple genes encode the same protein sequence (i.e. duplicate genes), or where a single gene produces multiple isoforms (often assigned distinct UniProt entries). Epigenetic genes were manually sorted into functionally defined families (i.e. histone acetyltransferases, histone deacetylases, etc.) based on previously characterized orthology and domain predictions (Supplementary File 6).

Prior to any filtering, 91.8% of the 28,896 proteins included in the *P. pacificus* annotation were either sorted into an OrthoFinder orthogroup or annotated with a Pfam domain by InterProScan (irrespective of annotated function); the remaining 8.2% presumably reflect lineage-specific genes and/or poor protein annotations.

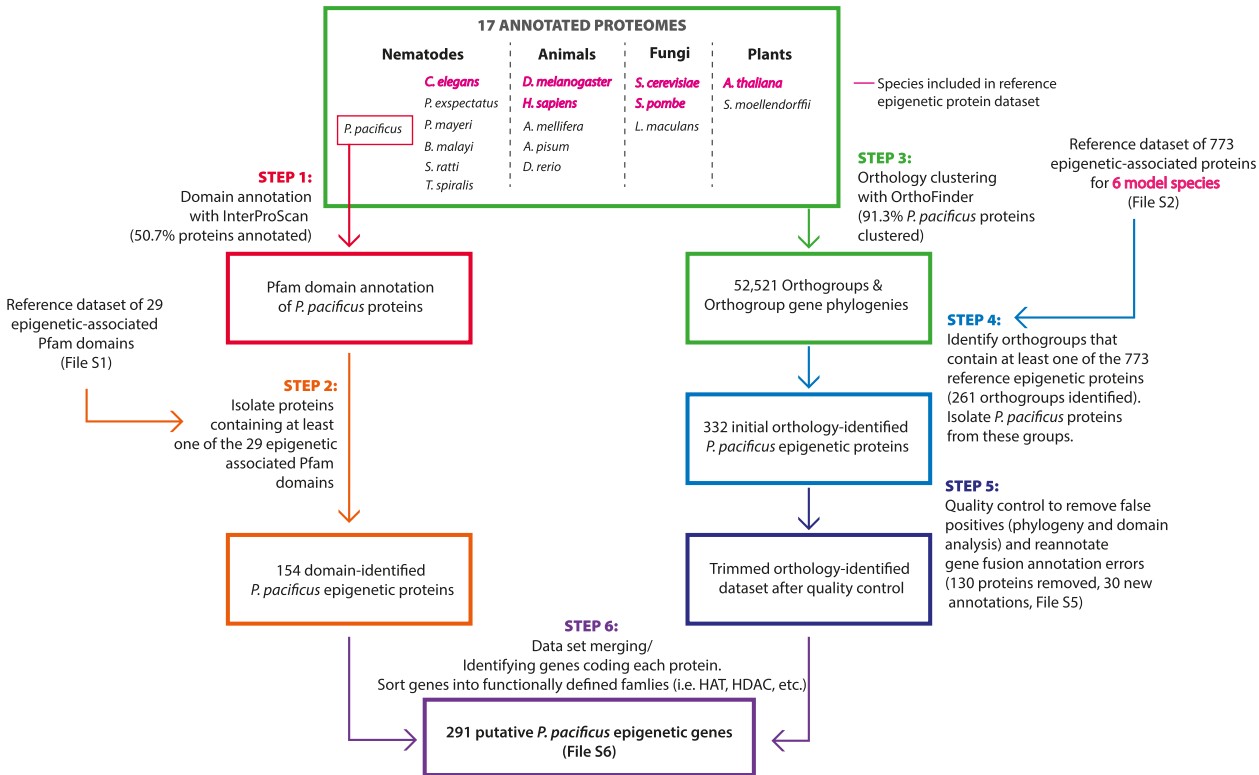

**Fig. 1.** Pipeline for identifying putative epigenetic genes in *Pristionchus pacificus*. (1) *Pristionchus pacificus* Pfam protein domains were annotated using InterProScan. (2) *Pristionchus pacificus* proteins containing at least one epigenetic-associated Pfam domain were identified. (3) All proteins from *P. pacificus* and 16 other species were clustered into orthogroups using OrthoFinder. (4) Orthogroups containing known epigenetic proteins from at least one of six reference model species were identified, and *P. pacificus* proteins from these were isolated. (5) This orthology-identified dataset was subjected to a series of quality control measures: analyzing orthogroups for phylogenetic correctness and consistent domain composition, and manual reannotation of gene fusion annotation errors. (6) The domain-identified and orthology-identified datasets were merged, and protein-coding genes were compiled into a final epigenetic gene dataset. Adapted from Pratx *et al.* 2018.

To validate our pipeline and to produce a comparative dataset to *P. pacificus*, the same workflow was used to identify *C. elegans*' epigenetic genes, with the modification of first removing all *C. elegans* proteins from the reference epigenetic-protein dataset (Supplementary Files 7–9). All data used during these analyses are included in Supplementary files. Supplementary File 1 contains the reference epigenetic Pfam domains; Supplementary File 2 contains the reference model-organism epigenetic proteins; Supplementary File 3 contains the InterProScan results for *P. pacificus* proteins; Supplementary File 4 contains the predicted Orthogroups; Supplementary File 5 contains information on manually reannotated genes; Supplementary File 6 contains predicted epigenetic genes for *P. pacificus*; Supplementary File 7 contains the InterProScan results for *C. elegans*; Supplementary File 8 contains predicted epigenetic genes for *C. elegans*.

## Orthology quality control

In step #5 of our pipeline, we performed three quality control measures:

1) We manually checked the placement of *P. pacificus* genes for phylogenetic correctness within each orthogroup by viewing the resolved gene tree output from OrthoFinder with Dendroscope version 3.8.3 (Huson and Scornavacca 2012). Unless containing an epigenetic-specific protein domain, *P. pacificus* genes were removed from the dataset if they (1) were a member of a tree or subtree that did not broadly recapitulate known phylogenetic relationships or (2) were

phylogenetically misplaced in an otherwise acceptable tree/subtree. Given that the gene trees were based on single gene alignments, and considering the broad phylogenetic scale involved, we did not require perfect recapitulation of established relationships; rather, we simply checked for clear and obvious departures suggestive of orthology group errors.

2) We performed a domain search on each protein within the candidate orthogroups using InterProScan and manually validated these with NCBI's CD-Search (Lu *et al.* 2020). For each orthogroup, we identified *P. pacificus* proteins with "outlier" domains that were not found in any other protein within the orthogroup. Such proteins were discarded if they also (1) did not contain any epigenetic-specific domains, (2) had no predicted epigenetic functions (from NCBI's CD-Search), or (3) were not closely related to other *P. pacificus* or model-organism epigenetic proteins (using the phylogenies from #1). In addition, we also excluded proteins with no predicted domains unless they were close orthologs of another *P. pacificus* or model-organism epigenetic protein.

3) We manually reannotated several gene fusion annotation errors in *P. pacificus* histone genes. These were cases where we detected the presence of multiple core histone domains within single gene annotation entries. Manual inspection of start/stop codon positions and existing Iso-seq reads (Werner *et al.* 2018) revealed these were in fact distinct histone genes, closely linked in tandem (Supplementary Fig. 6 and Supplementary File 5).

## Statistics on epigenetic gene counts

To compare the gene counts per epigenetic category between *P. pacificus* and *C. elegans,* we performed Fisher's exact tests using Monte Carlo simulation on the *C. elegans* and *P. pacificus* putative epigenetic gene count datasets (Table 1). Comparing the size of the gene families (i.e. Fisher's exact test on the $8 \times 2$ contingency table of family-by-species sums) suggested there might be a difference in specific epigenetic pathways ($P = 0.065$). However, we found no evidence for a difference between the two species at the subfamily level ($P = 0.941$, Fisher's exact test on the $33 \times 2$ contingency table of subfamily-by-species sums). We also performed $2 \times 2$ Fisher's exact tests on contingency tables of each protein family-level sum vs the sum of all others. This was done in R, version 4.4.2. The *P*-value significance thresholds were adjusted via Bonferroni correction to account for multiple testing.

## Sequence alignments and phylogenetic construction

All protein and nucleotide sequence alignments were created using the online version of MAFFT version 7 (Katoh *et al.* 2019). Orthogroup phylogenies were retrieved from OrthoFinder as described above. To construct all other phylogenies, protein sequence alignments were processed with ClipKIT (Steenwyk *et al.* 2020), and phylogenies were constructed using IQ-TREE 2 with the automated model testing setting and 1,000 bootstrap replicates (Minh *et al.* 2020). Phylogenies were graphed using either the R package "ggtree" (unrooted trees) or "phangorn" (midpoint rooted trees) (Schliep 2011; Yu *et al.* 2017). All tools were run using default settings unless otherwise indicated.

## Histone categorization and mapping

*Caenorhabditis elegans* canonical and variant H2A, H2B, H3, and H4 histone sequences have been previously characterized (Pettitt *et al.* 2002; Keall *et al.* 2007). We identified the *P. pacificus* canonical histone sequences as those genes with the best TBLASTN hit using the *C. elegans* canonical sequences as query—all other sequences were designated as variants. Ninety-five percent of *P. pacificus* and 98% of *C. elegans* canonical genes also contained a conserved hairpin sequence in the 3′ UTR, indicating they are replication-dependent (RD) (Supplementary Fig. 2b; Pettitt *et al.* 2002). Nucleotide sequence logos of the RD 3′ UTR hairpin sequence were plotted from nucleotide alignments of 100 bases immediately 3′ of all the stop codons of canonical histones using the "ggseqlogo" R package (Wagih 2017). Histone chromosomal positions were mapped using custom code written in R version 4.4.2 (https://github.com/audreybrown1/Brown-et-al.-2023-library, Supplementary Files 13 and 14). Histone clusters in *Caenorhabditis bovis, P. mayeri,* and *Allodiplogaster sudhausi* were identified by BLASTP of H4 as it is the most conserved histone with the fewest variants, and then manually searching for other histone genes in the vicinity (Stevens *et al.* 2020; Wighard *et al.* 2022; Prabh *et al.* 2018). Note, the genomes of *P. mayeri* and *A. sudhausi* are assembled from short-read Illumina sequences and are of poorer overall quality; thus, we are likely missing several histone gene clusters in these species. All genetic resources used are indicated in Supplementary Table 1, and Supplementary Files 13 and 14 contain the *P. pacificus* and *C. elegans* histone gene position data used to generate Fig. 2e and f.

## Histone liquid chromatography tandem-mass spectrometry (LC-MS/MS)

Histones from *P. pacificus* were acid-extracted following Werner *et al.* (2023). Briefly, cultures of *P. pacificus* were synchronized by bleach-NaOH, and eggs were aliquoted on to standard NGM-agar plates. Worm pellets (200–500 µl size) were collected and flash-frozen in liquid $N_2$ after 72 h, representing primarily adults. Crude nuclei preparations were made from worm pellets and then histones were extracted using sulfuric acid following Shechter *et al.* (2007). After extraction, 50–100 µg of histones were reduced with 1 mM DTT, alkylated with 50 mM ammonium bicarbonate/10 mM chloroacetamide and then digested with Arg-C, an endoprotease that cleaves polypeptides after Arg and Lys. Ten micrograms of Arg-C (Promega, cat. #V1881) was resuspended in 10 µl buffer (50 mM ammonium bicarbonate, 5 mM CaCl2, 2 mM EDTA) following the manufacturer's protocol. Resuspended Arg-C was used at 1:50 concentration (Arg-C:histones; 0.02 µg/µl) for in-solution digest of histones. To stop the digestion reaction, 10% trifluoroacetic acid was added to a final concentration of 0.5%.

Analysis of digested histones was done on an Easy-nLC 1200 system coupled to a QExactive HF-X mass spectrometer (both Thermo Fisher Scientific) as described in Werner *et al.* (2023). MS data were analyzed by MaxQuant software version 1.5.2.8 (Cox and Mann 2008) with integrated Andromeda search engine (Cox *et al.* 2011). For the detection of peptides cleaved by Arg-C, we selected proteolytic fragments with a maximum of two or three missed cleavages. The minimum peptide length was set to 5. Carbamidomethylation on cysteine was set as fixed modification in all processing of different combinations of variable modifications. These were mono-, di-, and trimethylation on lysine and arginine, phosphorylation on serine, threonine, and tyrosine, acetylation, crotonylation, butyrylation, hydroxybutyrylation, propionylation, and di-glycine on lysine, and O-GlcNacylation on serine and threonine. No more than five modifications were searched at the same time.

Data were mapped to the *P. pacificus* "El Paco" protein annotation version 1 (Rödelsperger *et al.* 2017), with a quality control threshold score >100 and posterior error probability (PEP) < 0.01. Initial maximum allowed mass tolerance was set to 4.5 parts per million (ppm) for precursor ions and 20 ppm for fragment ions. Peptide, protein and modification site identifications were reported at a false discovery rate of 0.01, estimated by the target/decoy approach (Elias and Gygi 2007).

## Prediction of *P. pacificus* histone PTM "writers" and "erasers"

*Pristionchus pacificus* orthologs of previously characterized *C. elegans* and human histone-modifying proteins were identified based on manual inspection of their previously generated orthogroup assignments and the associated orthogroup phylogenies. Human and *C. elegans* proteins known to "write" or "erase" specific acetyl and methyl modifications were identified from literature and database searches. The majority of histone PTM literature is on acetylation or methylation, partially because these are among the most abundant modifications, and partially because they correlate strongly with gene expression (Allis and Jenuwein 2016). Therefore, we chose to narrow our search to acetyltransferases, deacetylases, methyltransferases, and demethylases (Supplementary File 9). Supplementary File 9 contains all literature references used to generate Tables 2 and 3.

## PRC2 ortholog identification in *Pristionchus* nematodes

We used the genomes, protein sequences, and gene annotations of 10 diplogastrid nematode species to identify and analyze the conservation of the PRC2 complex within the *Pristionchus* lineage:

**Table 1.** Protein-coding epigenetic genes in *P. pacificus* and *C. elegans*.

| Family | Subfamily | *P. pacificus* | *C. elegans* | *p* value |
|---|---|---|---|---|
| HISTONE ACETYLTRANSFERASE | CBP/P300 | 12 | 7 | 0.170 |
| | GNAT | 15 | 11 | |
| | MYST | 7 | 4 | |
| | Other | 2 | 1 | |
| HISTONE DEACETYLASE | Class I | 4 | 3 | 1 |
| | Class II | 4 | 5 | |
| | Class III | 5 | 4 | |
| | Class IV | 1 | 1 | |
| HISTONE METHYLTRANSFERASE | PRMT | 12 | 3 | 0.052 |
| | SET | 45 | 31 | |
| | DOT1 | 6 | 6 | |
| | Other | 1 | 1 | |
| HISTONE DEMETHYLASE | JMJ | 20 | 12 | 0.399 |
| | LSD | 2 | 3 | |
| HISTONES | H1 | 5 | 9 | 0.004* |
| | H2A | 13 | 19 | |
| | H2B | 12 | 17 | |
| | H3 | 19 | 24 | |
| | H4 | 12 | 16 | |
| DNA METHYLATION | DNA methyltransferase | 1 | 0 | 1 |
| | ALKB | 5 | 5 | |
| | DAMT-1 | 1 | 1 | |
| NCRNA ASSOCIATED | Dicer | 1 | 1 | 0.609 |
| | RdRP | 5 | 4 | |
| | Helicases | 7 | 4 | |
| | Argonautes | 23 | 20 | |
| | Drosha | 1 | 1 | |
| | Microprocessor complex | 2 | 1 | |
| CHROMATIN REMODELERS | SWI/SNF | 2 | 2 | 1 |
| | ISWI | 2 | 1 | |
| | INO80 | 1 | 1 | |
| | CHD | 3 | 4 | |
| OTHERS | Histone kinase | 17 | 22 | 0.475 |
| | Histone phosphatase | 5 | 6 | |
| | Histone ubiquitin transferase | 9 | 7 | |
| | Histone deubiquitinase | 6 | 6 | |
| | Misc. | 3 | 2 | |
| TOTAL | | 291 | 265 | |

Table summary of epigenetic genes identified in *P. pacificus* and *C. elegans*. Values indicate number of genes recovered by our bioinformatic pipeline. Proteins were classified into families and subfamilies based on domain composition. *P*-values were calculated using Fisher's exact tests on 2 × 2 contingency tables of *P. pacificus* and *C. elegans* protein family sums vs the sum of the remaining epigenetic genes. *$P < 0.05$ (remains significant after Bonferroni correction).

*P. pacificus, P. exspectatus, Pristionchus arcanus, Pristionchus maxplancki, Pristionchus japonicus, P. mayeri, Pristionchus entomophagus, Pristionchus fissidentatus, Micoletzkya japonica,* and *Parapristionchus giblindavisi*. All genetic resources used for these species, including genome assemblies, transcriptomes, and RNA sequencing reads, are indicated in Supplementary Table 1 (Yoshida *et al.* 2023; Rödelsperger *et al.* 2017; Athanasouli *et al.* 2020; Werner *et al.* 2018; Prabh *et al.* 2018; Rödelsperger 2021). Genomes were viewed using IGV version 2.12.2 (Robinson *et al.* 2011).

We used BLAST and OrthoFinder to search for orthologs of the PRC2 proteins MES-2/EZH2 (*C. elegans* ID/Human ID), MES-3/SUZ12, and MES-6/EED in these 10 species. BLASTP was used to search predicted proteomes and TBLASTN was used to search the assembled transcriptomes, and genomes (and also in *P. pacificus,* the raw PacBio reads) of each nematode species, using both the human and *C. elegans* PRC2 sequences as query. For MES-2/EZH2 and MES-3/SUZ12, a significance threshold for BLASTP and TBLASTN of the transcriptome was set by plotting the e-values of the top hits (pooled across all 10 species) and then identifying where the hit e-values plateaued: based on this approach, we chose a threshold of 10e-29 (Supplementary Fig. 5a and b). As expected, TBLASTN to the whole genome/raw PacBio reads produced fewer and lower significance hits (results not shown), and therefore, a threshold of 10e-5 was used. We relaxed our search threshold to 10e-1 for all BLAST-based searches for MES-3/SUZ12;

however, even with this relaxed threshold we did not find any hits using either the *C. elegans* or human sequences as query. For OrthoFinder, we clustered the above-listed 10 species plus *C. elegans* and humans to search for PRC2 orthologs of human and *C. elegans* PRC2 components in each of the 10 diplogastrid nematodes: these orthogroups are contained within Supplementary File 10.

We also manually searched for *mes-2* and *mes-6* orthologs, and confirmed the annotations of previously identified orthologs, using formerly generated RNA-seq reads and positional information. For species in which we found an ortholog using BLAST or OrthoFinder, we isolated the genomic region spanning approximately 50,000–70,000 bases upstream and downstream and identified neighboring genes. Then, for species in which a *mes-2/mes-6* ortholog could not be found using BLAST or OrthoFinder, we searched for orthologs of neighboring genes to isolate the same genomic region. Each region was visually inspected in IGV alongside mixed-stage RNA-seq data (Rödelsperger *et al.* 2018) that was mapped to the genomes using the STAR sequence alignment software, version 2.7.10 (Dobin *et al.* 2013). Newly identified orthologs and gene annotations that did not match the RNA transcript data were manually reannotated (Supplementary File 5). Additionally, we used previously annotated repetitive sequences for *P. pacificus* to search the focal genomic regions for the presence of repetitive sequences indicative of transposons (Athanasouli and Rödelsperger 2022).

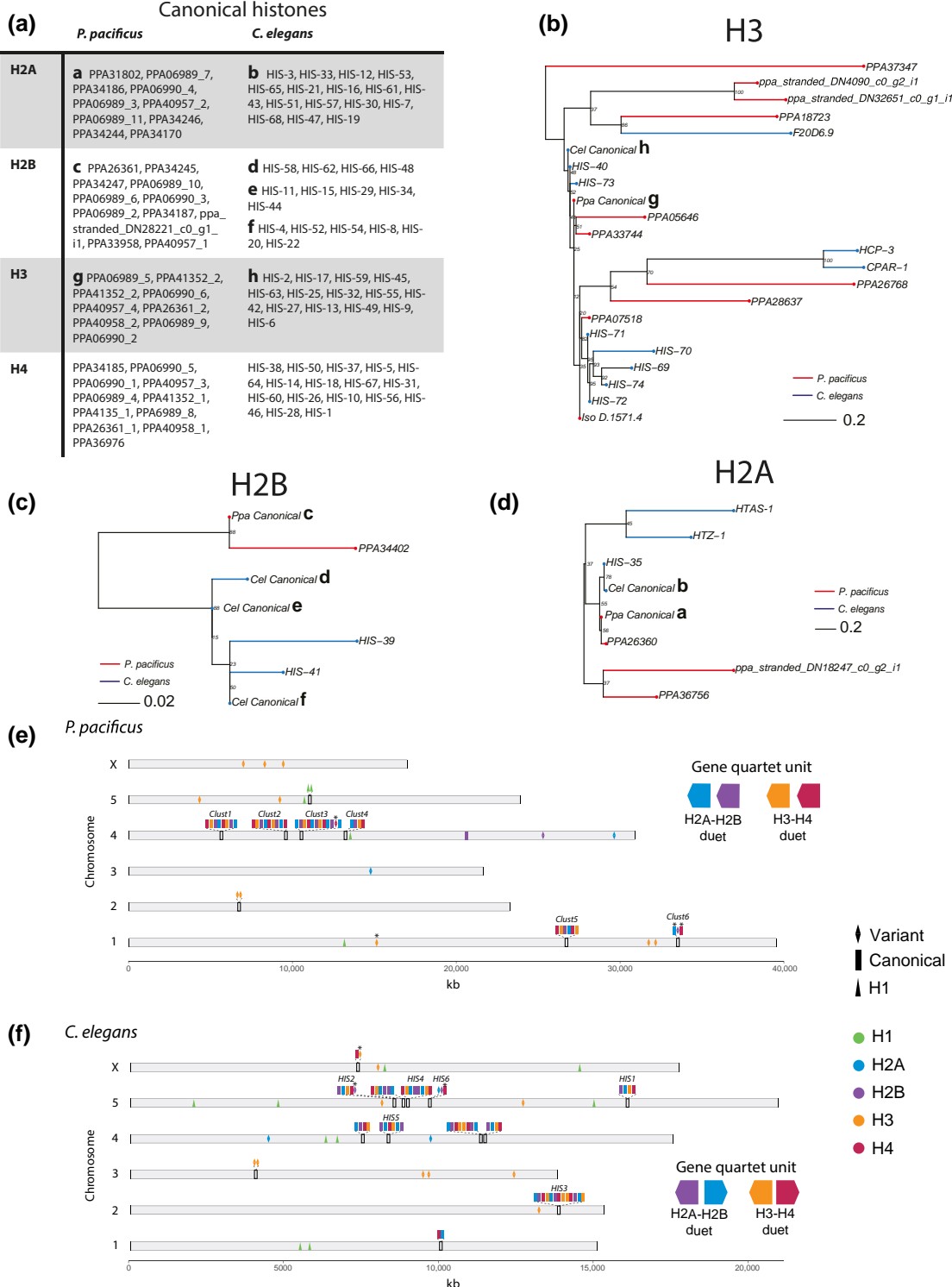

**Fig. 2.** Comparison of *P. pacificus* and *C. elegans* histones. a) Table of *P. pacificus* and *C. elegans* canonical histones. *P. pacificus* gene names ending with an underscore and number were manually reannotated (Supplementary File 5) b–d) H2A, H2B, and H3 phylogenies generated using maximum likelihood from aligned histone amino acid sequences and midpoint rooted. Branch length reflects the average number of amino acid substitutions per site. Canonical histones indicated with a lowercase letter, corresponding with the table in a). Note: *Pristionchus pacificus* H4 contains one variant not shown, PPA06989_1 e, f) Histone chromosomal positions. Asterisk (*) indicates cases where a variant is RD (containing a 3′ UTR hairpin associated with replication dependency) or a canonical gene is not RD (not containing a 3′ UTR hairpin). *C. elegans* clusters and names as described elsewhere (Roberts *et al.* 1987, 1989; Pettitt *et al.* 2002).

## Synteny analysis

We arranged each genomic region from the previous analysis based on the *Pristionchus* phylogeny (we excluded *M. japonica* and *P. giblindavisi* from this analysis due to short scaffold lengths). For each gene within this region, start and stop nucleotide positions were plotted using custom R code (https://github.com/audreybrown1/Brown-et-al.-2023-library). Genes were grouped according to their previously calculated orthogroups

**Table 2.** *Pristionchus pacificus* histone acetyltransferases and deacetylases.

| Family | Writer/eraser (*C. elegans*, *H. sapiens*) | Known modifications (present in *P. pacificus*) | *P. pacificus* ortholog |
|---|---|---|---|
| HAT | **pcaf-1,** KAT2A/GCN5 | H3K9/14ac | ppa_stranded_DN27027_c0_g1_i1 |
| HAT | **cbp-1, cbp-2, cbp-3,** CREBBP/P300 | H3K27ac, H3K9/14/18/23ac, H4K5/8/12/16ac | PPA40038, Contig11-snapTAU.521 |
| HAT | **mys-1,** KAT5 | H3K14ac, H4K5/8/12/16ac | PPA17126 |
| HAT | **mys-2, mys-4,** KAT7 | H3K14ac, H4K5/8/12ac | PPA37093, ppa_stranded_DN30216_c0_g1_i2, ppa_stranded_DN30317_c2_g1_i3 |
| HAT | **hat-1,** HAT1 | H4K5/12ac | ppa_stranded_DN13768_c0_g2_i1 |
| HAT | **eco-1,** ESCO-1 | – | ppa_stranded_DN25124_c0_g2_i1 |
| HAT | **taf-1,** TAF1 | – | ppa_stranded_DN29787_c1_g1_i2 |
| HAT | **elpc-3,** ELP3 | – | ppa_stranded_DN31509_c0_g3_i1 |
| HAT | NATD1 | – | PPA25554 |
| HAT | NAA60 | – | ppa_stranded_DN16451_c0_g1_i2 |
| HAT | SAT1, SAT2 | – | PPA04417, PPA05542, PPA29290, ppa_stranded_DN23261_c0_g1_i1, ppa_stranded_DN26404_c0_g1_i1 |
| HAT | NAA50 | – | PPA22455 |
| HAT | NAA40 | – | PPA34398, ppa_stranded_DN20546_c0_g2_i1 |
| HAT | **hda-1,** HDA1 | – | PPA17629, isopolya.1742.2 |
| HAT | **mys-3/lsy-12** | – | ppa_stranded_DN30317_c2_g1_i3 |
| HDAC | **hda-2,** HDA2 | – | ppa_stranded_DN28176_c1_g1_i1 |
| HDAC | **hda-3,** HDA3 | – | ppa_stranded_DN28546_c0_g1_i2 |
| HDAC | **hda-4,** HDA4 | – | PPA00049 |
| HDAC | **hda-10,** HDA10 | – | PPA35591 |
| HDAC | **hda-6,** HDA6 | – | ppa_stranded_DN30152_c3_g3_i1 |
| HDAC | **hda-5,** HDA5 | – | PPA35416 |
| HDAC | **sir-2.2, sir-2.3** | – | PPA17520 |
| HDAC | **sir-2.4** | – | PPA06876 |
| HDAC | SIR7 | H3K18ac | PPA34358 |
| HDAC | **sir-2.1** | H4K16ac | PPA18927, PPA35522 |
| HDAC | **hda-11,** HDA11 | – | PPA03988 |

*Pristionchus pacificus* orthologs of known *C. elegans* and human histone acetyltransferases (HATs) and histone deacetylases (HDACs), identified from orthogroup phylogenetic analysis. *C. elegans* genes are indicated in bold.

(Supplementary Files 11 and 12). Supplementary Files 11 and 12 contain all gene position data used to generate the *mes-2* and *mes-6* synteny plots (Fig. 5b and c).

## RNA-seq analysis

To analyze the expression of the PRC2 genes *mes-2* and *mes-6* across species, we obtained mixed-stage, paired-end RNA-seq data for *P. fissidentatus* (1 replicate), *P. entomophagus* (1 replicate), *P. mayeri* (1 replicate), *P. japonicus* (1 replicate), *P. maxplancki* (1 replicate), *P. arcanus* (5 replicates), *P. exspectatus* (5 replicates), and *P. pacificus* (5 replicates) from the European Nucleotide Archive (accession number PRJEB20959; Rödelsperger *et al.* 2018). Reads were aligned to the reference genomes using the STAR version 2.7.10 sequence alignment software, SAM output files were processed using Samtools version 1.16, and reads were quantified using htSeq-count version 2.0.2 with the nonunique none setting (Dobin *et al.* 2013; Anders *et al.* 2015; Danecek *et al.* 2021). Expression for each gene was quantified as Fragments Per Kilobase of transcript per Million mapped reads (FPKM):

$$\text{FPKM} = \frac{\text{total reads per gene}}{(\text{gene length in bp}) * (\text{total number of mapped reads})}(10^9)$$

## Immunofluorescence

Immunofluorescence was performed as previously described (Phillips *et al.* 2009). In brief, adult *P. pacificus* or *C. elegans* were picked into a drop of Egg buffer on a coverslip, with 0.1% Tween and 0.06% sodium azide. Worms were dissected and then fixed by adding a drop of Egg buffer with 1% formaldehyde. The coverslip containing the fixed, dissected worms was transferred onto HistoBond slides and frozen on dry ice for >1 min. The coverslip was quickly removed, leaving the worms on the HistoBond slide, and worms were washed in −20°C methanol for 1 min, then washed for 5 min 3× in room temperature PBS-Tween. Worms were blocked in Roche Block solution for 30 min and then incubated overnight in a 1:500 or 1:1,000 dilution of primary antibodies. After incubation, worms were washed for 10 min 3× with PBS-Tween and then incubated in a 1:500 secondary antibody dilution for 1½ h. Following secondary antibody incubation, worms were washed for 10 min each in PBS-Tween, PBS-Tween with 0.5 mg/ml DAPI, and again with PBS-Tween. Slides were mounted with N-propyl gallate-glycerol. Confocal images were taken using a Zeiss LSM 880 confocal microscope equipped with an AiryScan and using a 63× 1.4 NA oil immersion objective (n = 4 for *P. pacificus*, n = 2 for *C. elegans*). Maximum intensity projection confocal images are shown throughout and were processed using Zen Blue 3.0 and ImageJ. Primary antibodies used were Rabbit Tri-Methyl-Histone H3 (C36B11) (Cell Signaling, #9733) and Histone H3 (1B1B2) Mouse mAb (Cell Signaling, #14269). Secondary antibodies used were Goat bAb to Rb IgG Alexa Fluor 647 (Abcam, #150079) and Goat anti-Mouse IgG3 Cross-Adsorbed Secondary Antibody Alexa Fluor 594 (ThermoFisher, #A21155).

**Table 3.** *Pristionchus pacificus* histone methyltransferases and demethylases.

| Family | Writer/eraser (*C. elegans, H. sapiens*) | Known modifications (present in *P. pacificus*) | *P. pacificus* ortholog |
|---|---|---|---|
| HMT | **set-17,** PRDM9 | H4K4me1/2/3, H3K36me3 | ppa_stranded_DN30240_c0_g14_i5 |
| HMT | **set-30** | H3K4me1 | ppa_stranded_DN28741_c0_g1_i1 |
| HMT | **set-2,** SETD1A/B | H3Kme3, H3K4me1 | PPA00927 |
| HMT | **set-16** | H3K4me3 | ppa_stranded_DN28698_c0_g2_i1 |
| HMT | **met-2/,** SETDB1/2 | H3K9me2, H3K9me3 | ppa_stranded_DN31410_c4_g4_i1, PPA43093 |
| HMT | SUV39H1 | H3K9me3 | PPA41803, ppa_stranded_DN22002_c0_g2_i1 |
| HMT | **set-18** | H3K36me2 | isopolya.8103.2 |
| HMT | **mes-4** | H3K36me2 | PPA11725, PPA27330 |
| HMT | **lin-59,** ASH1L | H3K36me2, H3K36me3 | ppa_stranded_DN31241_c1_g7_i1 |
| HMT | **set-1,** SETD8 | H4K20me1 | PPA02950 |
| HMT | **set-4,** KMT5B | H4K20me2 | PPA04336 |
| HMT | **prmt-5,** PRMT5 | – | ppa_stranded_DN8606_c0_g1_i1 |
| HMT | **prmt-1,** PRMT1/8 | – | Contig232-snapTAU.1, PPA00689, PPA35131, PPA37361, PPA40548, PPA42284, PPA43471, ppa_stranded_DN31419_c0_g5_i2, PPA34076, PPA42322 |
| HMT | **prmt-7,** PRMT7 | – | ppa_stranded_DN25326_c0_g1_i2 |
| HMT | **met-1** | – | PPA05095, PPA01230 |
| HMT | **set-11** | – | PPA23400 |
| HMT | **set-23** | – | ppa_stranded_DN25446_c0_g1_i4 |
| HMT | **set-29,** SETD4 | – | PPA37266 |
| HMT | **set-10** | – | PPA18468, PPA18466 |
| HMT | **set-27,** SETD3 | – | PPA19833 |
| HMT | **set-3** | – | ppa_stranded_DN27277_c0_g1_i1 |
| HMT | **blmp-1,** PRDM1 | – | PPA04978 |
| HMT | **fib-1** | – | isototal.703.1 |
| HMT | **dot-1.1,** DOT1L | H3K79me1, H3K79me2 | PPA03461, PPA05175, PPA21945, ppa_stranded_DN25525_c0_g1_i2 |
| HDMT | **rbr-2** | H3K4me | ppa_stranded_DN20116_c0_g1_i1 |
| HDMT | **jmjd-2** | H3K9me, H3K36me | PPA01940 |
| HDMT | **utx-1** | – | PPA10459 |
| HDMT | **jmjd-4** | – | ppa_stranded_DN29473_c0_g3_i4 |
| HDMT | **psr-1** | – | PPA42591 |
| HDMT | **jmjc-1** | – | PPA05105 |
| HDMT | **spr-5** | H3K4me | ppa_stranded_DN16305_c0_g1_i1 |
| HDMT | **jhdm-1** | H3K9me2 | ppa_stranded_DN31380_c0_g4_i1, ppa_stranded_DN31380_c0_g4_i8 |
| HDMT | **jmjd-1.1/jmjd-1.2** | H3K9me2, H3K23me2, H3K27me2 | PPA195767 |

*Pristionchus pacificus* orthologs of known *C. elegans* and human histone methyltransferases (HMTs) and histone demethylases (HDMTs), identified from orthogroup phylogenetic analysis. *C. elegans* genes are indicated in bold.

## Results

### An inventory of *P. pacificus* and *C. elegans* epigenetic genes

We identified the "epigenetic toolkit" available to *P. pacificus* through a combination of orthology and domain predictions similar to an approach by Pratx *et al.* (2018) (Fig. 1). Broadly, proteins and protein domains that are linked to epigenetic processes in a diverse set of model systems were used to identify orthologs in our focal study system. Specifically, we assessed whether each *P. pacificus* protein (1) has domains that are found exclusively in epigenetic-function associated proteins and/or (2) is orthologous to proteins in model species with known epigenetic function. We focused on the enzymatic "writers" and "erasers" of DNA and histone modifications, noncoding RNA biogenesis, and chromatin remodeling pathways, as well as histone proteins themselves. We reasoned that these genes provide the clearest indication of presence/absence of epigenetic pathways: if the enzymes are present, we hypothesize that other critical members of these pathways, as well as their functional outputs, exist in *P. pacificus*. Hence, we did not consider noncatalytic members of multicomponent complexes, histone PTM "readers", or other upstream and downstream effectors of epigenetic modifications in this study.

To make domain and orthology assessments in *P. pacificus* proteins, we relied on reference datasets of known epigenetic proteins and protein domains from six model species (Supplementary Files 1 and 2). These were initially curated by Pratx *et al.* (2018), but were updated with an additional 83 proteins identified from further literature and database searches, found by searching for phrases such as "histone acetyltransferase activity," "DNA demethylase," "RNA-mediated gene silencing," etc. In total, there were 773 reference epigenetically associated proteins and 29 reference epigenetically associated domains (Supplementary Files 1 and 2). This approach allowed us to capture orthologs in a high-throughput manner and identify genes that may be otherwise missed in 1:1 BLAST comparisons due to gene loss, gain or divergence.

As input for orthology clustering, we used the proteomes of 17 species, including *P. pacificus* and six model species from our reference epigenetic-protein dataset. On average, 87% of genes per species were able to be sorted into an orthogroup. Interestingly, only 9% of orthogroups contained representatives from >50% of all species. Furthermore, 46% of all orthogroups contained representatives from only one species. A total of 261 orthogroups contained reference epigenetic proteins, and 128 of these contained one or more *P. pacificus* proteins (Supplementary File 4). Of the 133 "epigenetic" orthogroups that did not include *P. pacificus* proteins, 67.6% appear to be clade-specific, containing proteins from only one or two closely related species (Supplementary File 4).

We identified 332 *P. pacificus* proteins from the 128 *P. pacificus*-containing "epigenetic" orthogroups. 130 of these proteins were

filtered out from our dataset after manual curation (see methods). For example, orthogroups containing epigenetically associated kinases tended to be extremely large and often contained many other proteins with nonepigenetic functions. As a result, we only included kinases where a close phylogenetic relationship with a known model-organism histone kinase could be observed. Ultimately, we retained 100 epigenetic orthogroups with *P. pacificus* genes, 42 of which contained more than one *P. pacificus* gene (5 orthogroups contained *P. pacificus*-specific expansions, 30 contained deeper, prespeciation expansions, and 7 orthogroups contained both). In parallel, we performed a Pfam domain search on all *P. pacificus* proteins, and identified an additional 59 genes not represented in our orthology analysis, but with epigenetically associated protein domains. 30 more genes were identified from re-annotation of artificial gene fusions in the *P. pacificus* histone gene annotations (Supplementary File 5). The final dataset includes 291 *P. pacificus* epigenetic-function genes (Table 1 and Supplementary File 6).

We also annotated putative *C. elegans* epigenetic genes to produce a comparative dataset (but with the prior removal of all *C. elegans* genes from our reference datasets; 160 total genes, 105 unique protein sequences). Here, our pipeline and quality control yielded a final dataset of 265 *C. elegans* epigenetic-associated genes (Table 1 and Supplementary File 8). We recovered 156 of the 160 (97.5%) reference *C. elegans* genes we had removed before applying the pipeline (Supplementary File 8). The four unrecovered reference genes each belong to a *C. elegans*-specific orthogroup and lack distinct epigenetic domains, which is presumably why we could not identify them from our approach. We also identified 109 additional putative epigenetic genes not previously included in the *C. elegans* reference set (Supplementary File 8). Upon closer inspection, 62 were annotated in UniProt as having epigenetic functions. Therefore, these likely represent bona fide epigenetic genes that were previously overlooked when assembling reference datasets. The remaining 47 had no explicit epigenetic annotations and may represent the result of duplication and divergence of other epigenetic-associated genes, or genes where epigenetic function has not been explicitly experimentally characterized. As an example, we found that in UniProt, the *C. elegans* gene ZK1098.11 is named N-acetyltransferase domain-containing protein 1 (NATD1), presumably because it is an ortholog of a human protein with the same name (Supplementary File 4). Our analysis revealed that both ZK1098.11 and human NATD1 are orthologs of *A. thaliana* protein At1g77540 (Supplementary File 4), which possesses in vitro acetyltransferase activity toward both H3 and H4 (Tyler *et al.* 2006). Taken together, these results suggest that combining protein domain queries with orthology can better identify epigenetic functions in proteins than either method alone, leading to the recovery of more genes than those currently annotated in databases such as UniProt (even in well-characterized organisms such as *C. elegans*).

Next, we compared the two focal species nematode datasets. We manually sorted genes from the *P. pacificus* and *C elegans* datasets into protein families based on their domain or subfamily composition (Table 1). Phylogenetic analysis of these families shows that within each, there are several instances of one-to-one orthology between *P. pacificus* and *C. elegans* (Supplementary Fig. 1). However, we also observed multiple instances where this is not the case. For example, the SET-containing *P. pacificus* proteins Contig47-snapTAU.201, Iso_D.594.1, PPA40945, and ppa_stranded_DN29031_c0_g1_i1 form a *P. pacificus*-specific clade when visualized in a phylogeny of *P. pacificus* and *C. elegans* SET

proteins (Supplementary Fig. 1). While these proteins have no apparent *C. elegans* ortholog, they are members of an orthogroup containing a reference *A. thaliana* protein SUVH10 (Supplementary File 4)—highlighting the utility of including species beyond *C. elegans* to identify epigenetic proteins in *P. pacificus*.

*Caenorhabditis elegans* and *P. pacificus* harbor several differences in epigenetic-protein composition at the subfamily level (Table 1). First, the two species both have at least one member of each subfamily, apart from the presence of a DNA methyltransferase (DNMT) in *P. pacificus*. These results are consistent with earlier reports establishing that *P. pacificus* contains a DNMT-2 like gene (Gutierrez and Sommer 2004). The majority of studied nematodes notoriously lack 5mC DNA methylation and have correspondingly lost most of the genes responsible for establishing and maintaining DNA methylation (Simpson *et al.* 1986; Engelhardt *et al.* 2022). This is also believed to be the case in *P. pacificus*: despite the presence of a DNMT-like gene, previous studies have shown that DNA methylation is not a genome wide phenomenon in *P. pacificus* (Gutierrez and Sommer 2004). Second, the PRMT gene family has the largest difference in gene count, with *P. pacificus* containing 4× as many PRMT genes as *C. elegans*. To further explore differences at the gene family level, we also compared the size of each individual family against the sum of all other genes. This analysis indicated a significant difference in the number of histone genes between species: *C. elegans* has a disproportionately large histone complement compared to *P. pacificus* ($P = 0.004$; Table 1). Indeed, this pattern appears to be consistent across each histone subfamily—*C. elegans* contains more copies of H1, H2A, H2B, H3, and H4 than *P. pacificus*.

## *Pristionchus pacificus* and *C. elegans* histone gene clusters are deeply diverged

In metazoans, histone genes have repeatedly undergone serial duplication events, and duplicate copies of H2A, H2B, H3, and H4 histones can be found clustered in groups and distributed throughout the genome many times over (Amatori *et al.* 2021). These clusters tend to encode "canonical" histones. Canonical histones are nearly identical at the amino acid level and are RD, with expression restricted to S-phase when the nucleosomes are doubled after DNA replication (Amatori *et al.* 2021). By contrast, "variant" histones often reside outside of these repeated arrays, have slightly modified amino acid sequences, and are expressed throughout the cell cycle. After finding that *C. elegans* contains a larger histone gene compliment than *P. pacificus*, we asked whether differences in the number of canonical or variant histones drove this variation. To answer this question, we examined the differences in canonical and variant histones between the two species using unrooted phylogenies of all *P. pacificus* and *C. elegans* histones generated from amino acid sequence alignments (Fig. 2a–d and Supplementary Fig. 2c). Within the core nucleosome histones (H2A, H2B, H3, and H4), *C. elegans* has 61 canonical genes and 15 variants, while *P. pacificus* has 41 canonical and 15 variants. Therefore, canonical histones appear to be driving the difference in histone gene number between the two species rather than histone variants. Answering this question also allowed us to observe additional differences between *C. elegans* and *P. pacificus* canonical histone sequences. First, we observed that *P. pacificus*' and *C. elegans*' canonical histones (except H4) differ slightly in amino acid sequence: H2A differs by seven amino acids, H3 by three amino acids, and H2B by 12–30 amino acids (*C. elegans* contains multiple canonical H2B sequences; Supplementary Fig. 2a). This follows a known pattern that H2A and H2B are less conserved

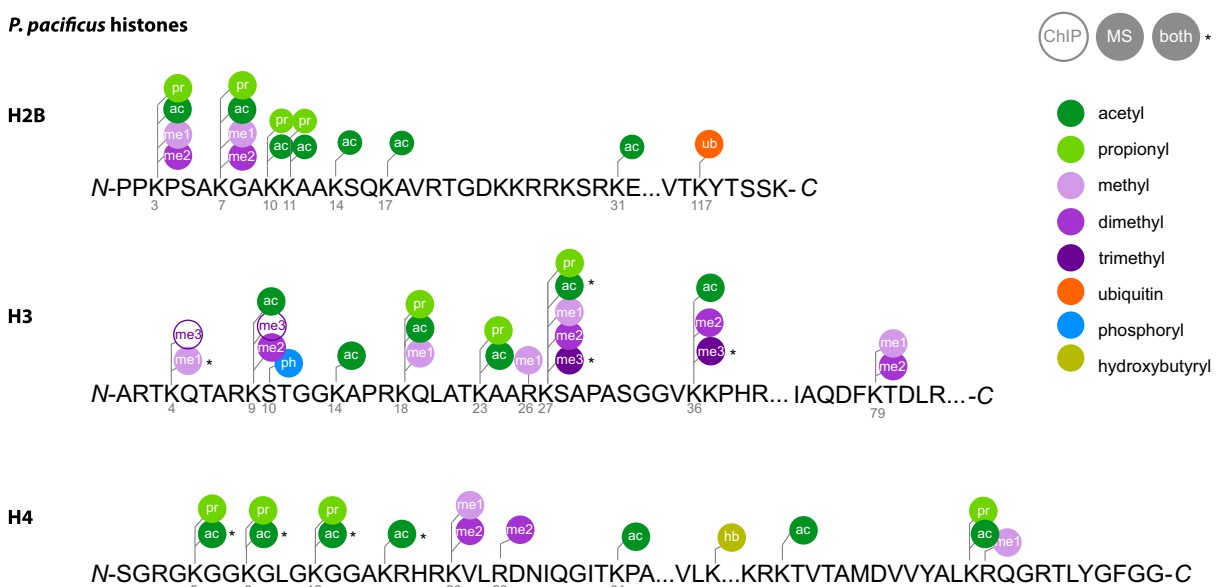

**Fig. 3.** *Pristionchus pacificus* histone PTMs. Summary of all PTMs detected in histones extracted from adult *P. pacificus* pellets 72 h postsynchronization. PTMs were detected with LC-MS/MS (PEP score <0.01; this study), ChIP-seq (Werner *et al.* 2018, 2023), or both.

at the amino acid sequence level than H3 and H4 (Thatcher and Gorovsky 1994; Raman *et al.* 2022).

In *C. elegans,* 11 primary histone gene clusters comprised of "quartet" units—tandem repeats of canonical H2B, H2A, H3, and H4—have been previously characterized (Roberts *et al.* 1987; Pettitt *et al.* 2002). Therefore, we hypothesized that histone cluster duplications, deletions, or other genomic rearrangements may explain the differences in the number of canonical histone genes between the two species. Mapping both *P. pacificus* and *C. elegans* histones to their respective chromosomal position revealed that *P. pacificus* contains six canonical histone gene clusters (Fig. 2e and f). Excluding the replication-independent and largely variant containing *C. elegans* HIS6 cluster, and a homologous *P. pacificus* cluster we called Clust6, each *P. pacificus* cluster is comprised of quartet gene repeats like *C. elegans*. However, the two species contain different quartet units: the *P. pacificus* quartet typically contains genes in the order H2A, H2B, H3, and H4 (with an exception in Clust3) all oriented in the same direction (i.e. head-to-tail), whereas in *C. elegans* the histone genes are typically found in the order H2A, H2B, H4, H3 (with an exception in the first chromosome 4 cluster), with H2A and H4 oriented opposite of H2B and H3 (i.e. head-to-head). We found no *C. elegans* clusters containing the *P. pacificus* quartet unit or vice versa. Finally, we asked how conserved this pattern is—do other *Pristionchus* or *Caenorhabditis* species contain similar quartet orders and orientations? We used BLAST to identify histone clusters in *C. bovis,* *P. mayeri,* and the non-*Pristionchus* diplogastrid *A. sudhausi* (Supplementary Fig. 3). Within these related species, there is variability in the order of the histone genes within a quartet. However, within each quartet, the H2A–H2B and H3–H4 "duet" orientation is conserved. All *C. bovis* clusters contain H2A–H2B and H3–H4 duets where the genes are in a head-to-head orientation as in *C. elegans*. By contrast, *P. mayeri* and *A. sudhausi* only contain head-to-tail oriented duets as seen in *P. pacificus*. Thus, the order of duets is more evolutionarily labile than the orientation. These data also suggest that there is a deep divergence of the duplication events that formed the *Pristionchus* and *Caenorhabditis* histone gene clusters. We propose that *P. pacificus* and *C. elegans* histone clusters underwent lineage-specific expansions after the duplication of unique duet units. The difference in the number of canonical genes between the two species is likely a result of this independent cluster evolution.

## Characterization of *P. pacificus* histone PTMs, writers, and erasers

We were interested in defining the histone PTMs present in *P. pacificus* to pair with our putative epigenetic gene dataset. Previously, 11 histone PTMs (H3K4me1/3, H3K9me3, H3K9ac, H3K27me3, H3K27ac, H3K36me3, H4K5ac, H4K8ac, H4K12ac, and H4K16ac) have been observed using ChIP-seq and/or western blot (Werner *et al.* 2018, 2023; Serobyan *et al.* 2016). However, known PTMs are limited to the antibody of choice, and antibody off-target binding can generate misleading interpretations (Grzybowski *et al.* 2015). To better characterize the suite of histone PTMs in *P. pacificus*, we performed LC-MS/MS on histones extracted from adult worms. Using this more sensitive and comparatively unbiased method, we detected an additional 45 acetylation, methylation, ubiquitination, phosphorylation, propionylation, and hydroxybutyrylation sites on H2B, H3, and H4 (Fig. 3 and Supplementary Fig. 4). Histone acetylation, methylation, ubiquitination, and phosphorylation are well characterized across species; however, histone propionylation and hydroxybutyrylation have only recently been identified through targeted mass spectrometry (Kebede *et al.* 2017; Zhou *et al.* 2022). We were unable to detect any modifications on H2A in *P. pacificus*. H2A PTMs are less well characterized and thought to be less common than those on their other histone counterparts (Beck *et al.* 2006). One notable exception is H2A mono-ubiquitination (at lysine K119 in mammals, K120 in *C. elegans*), previously reported in *C. elegans* embryos (Samson *et al.* 2014). From our data, we cannot conclude whether this mark or any other H2A modifications are necessarily absent in *P. pacificus*: they may be present at levels below our detection threshold or only detectable at different developmental time points. Notwithstanding potential false negatives, this dataset provides a biochemical inventory of histone PTMs in *P. pacificus* that can be used to complement our bioinformatic inventory of genes.

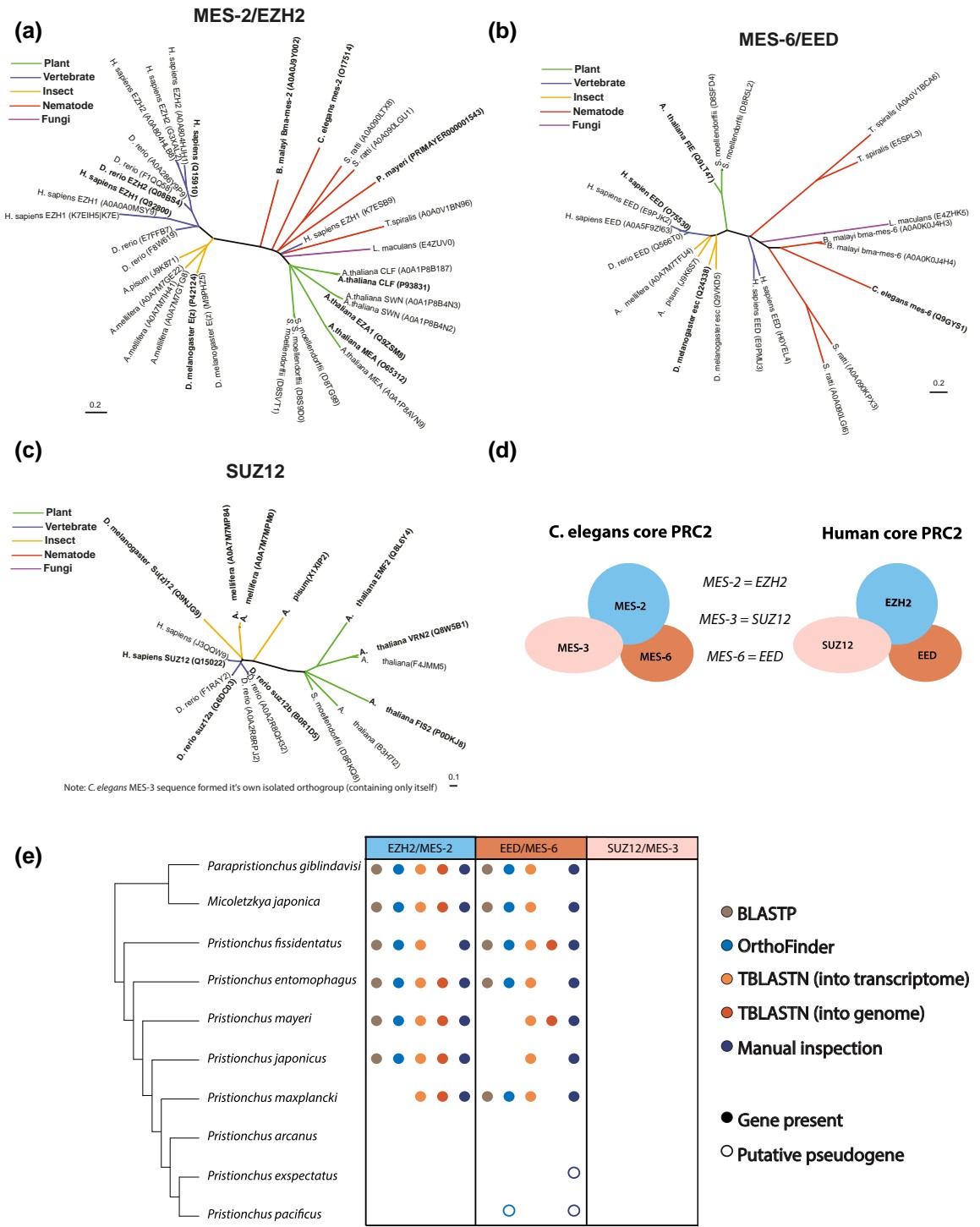

**Fig. 4.** The PRC2 methyltransferase is not conserved in Pristionchus. a–c) MES-2/EZH2, MES-6/EED, and SUZ12 orthogroups were generated using the pipeline illustrated in Fig. 1 (*C. elegans* MES-3 formed an isolated orthogroup containing only itself). UniProt accession numbers are listed in parentheses. For any nearly identical UniProt entries for the same protein (reflecting either separate protein isoforms or database redundancy), the primary (verified) UniProt entry is in bold. d) Schematic of human and *C. elegans* PRC2 orthologs. e) Identification of PRC2 component orthologs in the *Pristionchus* phylogeny via BLASTP (e < 10E-29); Orthology clustering via OrthoFinder of the 10 listed nematodes, *C. elegans*, and *H. sapiens*; TBLASTN into the transcriptome (e < 10E-29); TBLASTN into the genome (e < 10E-5); and (in cases where the genomic loci of interest could be identified), manual inspection of mapped RNA-seq reads.

We then asked whether *P. pacificus* contains orthologs of human and *C. elegans* acetyltransferases, deacetylases, methyltransferases, and demethylases to pair with our histone LC-MS/MS data, as these are the most well-characterized histone-modifying enzymes involved in gene regulation. To this end, we used our orthology dataset and orthogroup phylogenies to predict *P. pacificus* orthologs of human and *C. elegans* enzymes. From this analysis, we could predict 87 *P. pacificus* orthologs of previously characterized human and *C. elegans* histone-modifying enzymes (Tables 2 and 3). Twenty-one of the *C. elegans* or human proteins

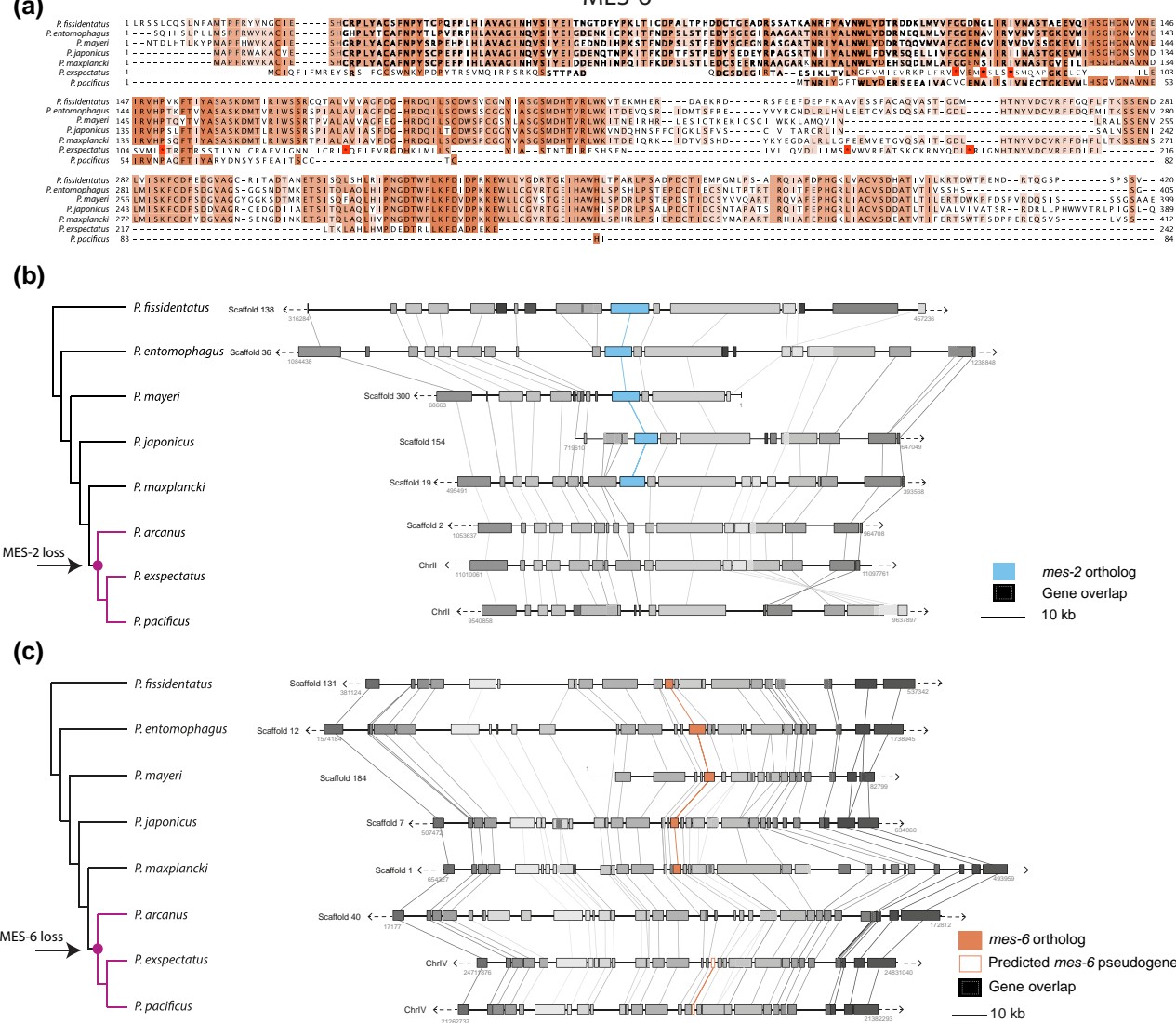

**Fig. 5.** *Pristionchus mes-2* and *mes-6* deletion and pseudogenization. a) Protein sequence alignment of MES-6 orthologs in *Pristionchus* nematodes, showing that the *P. exspectatus* ortholog contains many stop codons (*) and additionally departs from the consensus, and the *P. pacificus* ortholog is truncated. b, c) Synteny analysis of the genomic region surrounding *mes-2* and *mes-6* in *Pristionchus* nematodes. Genes are grouped according to orthogroup membership.

have known preferred substrates that have been detected in *P. pacificus* either from mass spectrometry (this study) or antibody-based methods (Table 2; Supplementary File 9; Werner *et al.* 2018; Werner *et al.* 2023). In combination, these data establish the repertoire of histone PTMs present in *P. pacificus* and provides candidates for the writers and erasers of specific marks.

## The methyltransferase complex PRC2 was lost in *P. pacificus* and related *Pristionchus* nematodes

From our "epigenetic toolkits" in *P. pacificus* and *C. elegans,* we noted many instances of species-specific gene gains and losses (Table 1 and Supplementary Fig. 1). Of these, we were surprised to find that *P. pacificus* was missing an ortholog of MES-2/EZH2 (*C. elegans* ID/human ID) (Table 3, Fig. 4a). MES-2/EZH2 is the enzymatic component of the highly conserved PRC2 complex, which deposits H3K27me3 to provide maintenance of gene repression (Ahringer and Gasser 2018). The PRC2 complex is the only known eukaryotic H3K27me3 catalyst in vivo; a SET protein in the PBCV-1 virus is the only other enzyme currently reported to catalyze H3K27me3, and G9a has been reported to catalyze H3K27me1/2 (Mujtaba *et al.*

2008; Wu *et al.* 2011). Manipulating EZH2 and/or H3K27me3 disrupts cell lineage specification, mammalian X chromosome inactivation and transposon silencing (Yuzyuk *et al.* 2009; Patel *et al.* 2012; Walter *et al.* 2016; Chen and Zhang 2020; Loh and Veenstra 2022). H3K27me3 is also one of the few histone modifications that has been shown to be propagated meiotically and/or mitotically, making it a true epigenetic information carrier (Margueron and Reinberg 2011; Gaydos *et al.* 2014; Oksuz *et al.* 2018; Kaneshiro *et al.* 2022). In the only eukaryotes for which PRC2 is known to have been lost—the unicellular yeasts *S. cerevisiae* and *S. pombe*—the H3K27me3 mark is also absent (Margueron and Reinberg 2011). However, H3K27me3 has been previously detected in *P. pacificus* with ChIP (Werner *et al.* 2023), and our new analysis confirmed this finding with LC-MS/MS (Fig. 3 and Supplementary Fig. 4). The presence of H3K27me3—but the seeming absence of its writer MES-2—compelled us to investigate further.

Two *P. pacificus* proteins (PPA10329 and PPA06844) cluster with *C. elegans* MES-2 from a phylogeny of SET domain-containing proteins (Supplementary Fig. 1a). However, this branch is supported by a low bootstrap (54), and these proteins are not present in the

MES-2 orthogroup phylogeny (Fig. 4a). Moreover, manual inspection revealed that homology to MES-2 was essentially limited to the SET domain. Though the *P. pacificus* genome is high quality (BUSCO score of 92.6; Rödelsperger *et al.* 2017), it is also possible that *mes-2* was integrated into a nonassembled part of the *P. pacificus* genome. To account for this scenario, we also performed tBLASTn of raw PacBio reads for orthologs of MES-2, but could not recover any hits. Thus, neither our bioinformatic pipeline nor a targeted analysis retrieved the catalytic subunit of the PRC2 complex in *P. pacificus*.

In both humans and *C. elegans*, the PRC2 complex contains three core proteins, each required for catalytic activity: the enzymatic component MES-2/EZH2 and two cofactors MES-3/SUZ12 and MES-6/EED (Fig. 4d) (Jiao and Liu 2015; Ahringer and Gasser 2018; Snel *et al.* 2022). We first examined our previous orthogroup phylogenies for each of these three proteins to see if we could identify orthologs in *P. pacificus*, as well as in the related *Pristionchus* nematodes *P. exspectatus* and *P. mayeri* (Fig. 4a–c). We only found a *P. pacificus* ortholog of MES-6 that is highly truncated, likely a nonfunctional pseudogene, and a *P. mayeri* ortholog of MES-2. *C. elegans* MES-3 formed an isolated orthogroup, separate from all other SUZ12 orthologs (consistent with previous reports that MES-3 and SUZ12 are highly diverged orthologs; Snel *et al.* 2022); neither orthogroup included any *Pristionchus* species. Based on these results, we wondered if PRC2 was lost in multiple *Pristionchus* nematodes.

To further expand our evolutionary analysis of the apparent loss of PRC2 in the *Pristionchus* lineage, we searched for MES-2, MES-6, and MES-3 orthologs in 10 related diplogastrid nematodes which form a ladder-like phylogeny with *P. pacificus* (Prabh *et al.* 2018), allowing us to connect the presence/absence of PRC2 components to specific branch points. We searched for PRC2 orthologs initially using four different methods: BLASTP, TBLASTN of the genome, TBLASTN of the transcriptome, and OrthoFinder analysis of the 10 diplogastrid nematode proteomes, plus those of *C. elegans*, and humans (Fig. 4e). Both BLASTP and TBLASTN consistently returned slightly more significant results for searches using the human sequences as the search query rather than *C. elegans* (Supplementary Fig. 5a and b). In addition to the previously identified *P. mayeri* MES-2 ortholog, we identified MES-2 and MES-6 orthologs in *P. giblindavisi, M. japonica, P. fissidentatus, P. entomophagus, P. mayeri,* and *P. japonicus*. We also manually curated genome alignments using RNA-seq data (Rödelsperger *et al.* 2018) to verify gene annotations and search for unannotated orthologs (Fig. 4e). For species with no identifiable *mes-2/mes-6* ortholog, neighboring genes were used to identify the corresponding genomic region. From this analysis, we discovered a putative *mes-6* transcript in *P. exspectatus*, albeit one that contained several stop codons (Fig. 5a). This analysis also revealed several *mes-2* and *mes-6* gene annotation errors in *P. fissidentatus, P. entomophagus,* and *P. japonicus*; reannotated versions of these genes were used in all further analysis (Supplementary Fig. 6 and Supplementary File 5). No *mes-3* orthologs could be found in any of the diplogastrids examined by any method—despite relaxing the BLAST threshold to 10e-1—indicating that this gene is either absent, or so diverged from the human and *C. elegans* sequences that these methods could not identify it. The combination of all approaches point to *mes-2* and *mes-6* being lost in the last common ancestor of *P. arcanus* and *P. pacificus*.

## Mes-2 and *mes-6* were lost by different evolutionary mechanisms

Pseudogenes often display decreased gene expression and sequence features such as premature stop codons and truncations

(Zhang and Gerstein 2004). Truncation of the predicted *mes-6* open-reading frame from *P. pacificus* and multiple stop codons in the predicted *mes-6* transcripts from its sister species *P. exspectatus* indicated pseudogenization (Fig. 5a). However, all *mes-6* genes, including the predicted pseudogenes, displayed expression by RNA-seq (Supplementary Fig. 7b). Premature stop codons or gene truncations were not observed in orthologs of *mes-2*: we either found a presumably functional full-length coding sequence or complete absence. Additionally, all existing *mes-2* orthologs exhibited expression by RNA-seq (Supplementary Fig. 7a). Thus, at least at this time, there is no apparent signature of decreased expression of *mes-2* along the *Pristionchus* phylogeny that predated its loss in some lineages.

The absence of *mes-2* orthologs (including pseudogenes) could be explained by rapid sequence evolution rendering pseudogenized *mes-2* orthologs unrecognizable, or it could be that the locus has been lost from the genome entirely. We asked whether we could find any indication of large genomic rearrangements that could be responsible for the sudden absence of *mes-2* orthologs. To this end, we analyzed the local synteny of the genomic region surrounding *mes-2* across the *Pristionchus* phylogeny (Fig. 5b). From this analysis, we determined that the genomic region surrounding *mes-2* is conserved across the phylogeny: even though *mes-2* is absent in *P. pacificus, P. exspectatus,* and *P. arcanus,* the positions of neighboring genes are relatively fixed. Additionally, the close proximity of the neighboring genes suggests that the *mes-2*-containing region may have been lost entirely in the common ancestor of *P. pacificus* and *P. arcanus*. We also repeated this analysis for *mes-6* and similarly found that the structure of this region is also well conserved across these nematodes, with pseudogenes replacing functional *mes-6* in *P. exspectatus* and *P. pacificus* (Fig. 5c). Finally, we asked whether there were any repetitive sequences indicative of transposon activity (which have been previously annotated in *P. pacificus*; Athanasouli and Rödelsperger 2022) that could explain *mes-2*'s loss. However, we did not detect any of these sequences at the *mes-2* locus in *P. pacificus*. Taken together, functional *mes-2* and *mes-6* genes appear to have been lost in the common ancestor of *P. arcanus* and *P. pacificus* but by different evolutionary mechanisms. *Mes-2* appears to have been lost via a specific deletion event, while *mes-6* has undergone a pseudogenization process, leaving behind gene remnants in *P. pacificus* and *P. exspectatus*.

## *Pristionchus pacificus* retains H3K27me3 in both its germline and somatic tissue

In *C. elegans*, H3K27me3 is present in essentially all body tissues, yet PRC2 is only essential for maintaining germline H3K27me3 (Bender *et al.* 2004). H3K27me3 in the *C. elegans* germline is enriched on the X chromosome and is responsible for maintaining X chromosome repression during germline development; loss of any PRC2 component causes loss of all germline H3K27me3 and a maternal effect sterile phenotype (Capowski *et al.* 1991; Bender *et al.* 2004; Strome *et al.* 2014). Because *P. pacificus* appears to have lost PRC2, we asked whether it has correspondingly lost H3K27me3 in its germline but not somatic tissue. In this scenario, the lack of PRC2 in *P. pacificus* would indicate species-specific differences in germline maintenance and explain the detection of H3K27me3 as present exclusively in the soma.

We dissected the gonad and intestine (somatic tissue) from wild-type adult *P. pacificus* and *C. elegans* worms and stained them with antibodies specific for H3K27me3, H3, and with DAPI to stain nuclei (Fig. 6 and Supplementary Fig. 8). As expected, *C. elegans* shows robust H3K27me3 staining in both the gonad

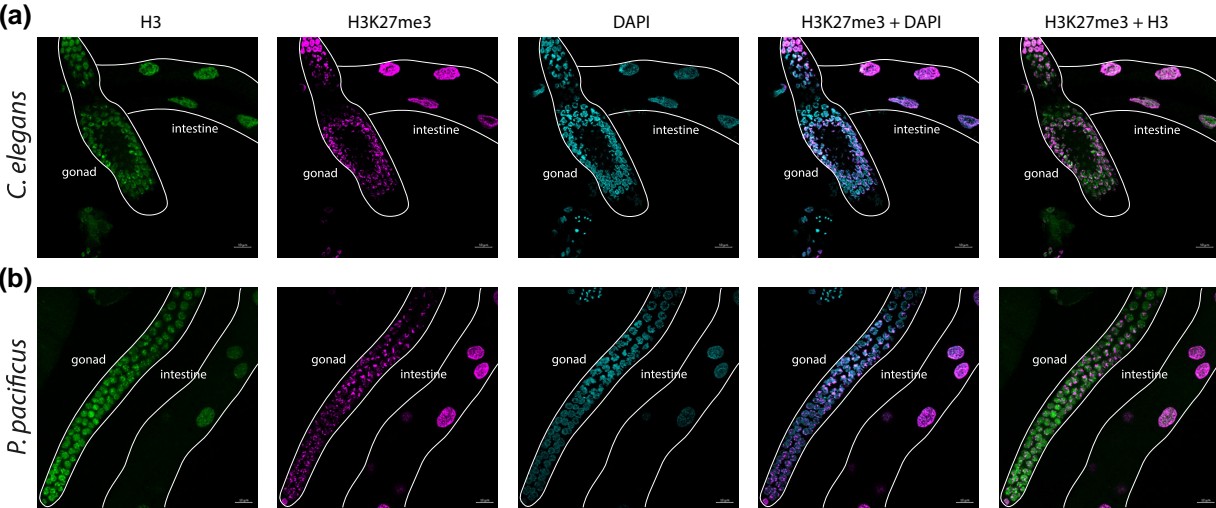

**Fig. 6.** *Pristionchus pacificus* contains H3K27me3 in both the soma and the gonad. Immunofluorescence staining of dissected (a) *C. elegans* and (b) *P. pacificus* gonad and intestine. n = 4 for *P. pacificus* and 2 for *C. elegans*. See methods for details.

and intestine. Interestingly, *P. pacificus* also shows H3K27me3 staining in both tissues. Furthermore, H3K27me3 does not appear to cover all chromatin marked by H3, suggesting it is enriched on specific chromosomes (Supplementary Fig. 8f). Therefore, H3K27me3 is retained in the *P. pacificus* gonad, potentially for X chromosome silencing (as in *C. elegans*), despite the loss of PRC2.

## Discussion

Recent pharmacological experiments indicate that epigenetic information carriers, particularly histone PTMs, regulate mouth-form plasticity in the evo-devo model system *P. pacificus* (Werner *et al.* 2023). However, validating this result and connecting it to the action of individual genes requires functional information on the repertoire of epigenetic writers and erasers encoded within the *P. pacificus* genome. To address this shortcoming, we have created an inventory of putative epigenetic genes in *P. pacificus,* as well as in *C. elegans,* using a domain and orthology-informed pipeline. These datasets provide a foundation for experimental analysis of plasticity and gene regulation in both *P. pacificus* and *C. elegans*. Additionally, a comparison of the epigenetic toolkits between both model nematodes highlights unexpected facets of evolutionary divergence.

First, there was a significant difference in histone gene composition. In both species, core "quartet units" of canonical histone genes have multiplied throughout the genome to form clusters (Roberts *et al.* 1987; Pettitt *et al.* 2002). Despite the high conservation of histone genes themselves, the *P. pacificus* and *C. elegans* histone clusters contain quartets comprised of distinct duets (in terms of gene order and orientations), with no instances of the *P. pacificus* duets being found in *C. elegans* or vice versa. We also found that the respective *C. elegans* and *P. pacificus* histone gene duets are conserved among each's close relatives (in *C. bovis,* for *C. elegans;* and in *P. mayeri* and *A. sudhausi,* for *P. pacificus*). These findings are consistent with a deep divergence of *P. pacificus* and *C. elegans* (estimated at 80–200 million years; Howard *et al.* 2022). We propose that the observed differences in the total canonical histone genes between *P. pacificus* and *C. elegans* are due to independent histone gene cluster formation—indicated by the duplication of distinct gene duets in each species. Going forward, it will be interesting to see when each histone gene cluster arose,

which may help to resolve relative nematode phylogenetic positions. Recent progress in nematode genomics is providing the raw data needed for such analyses (Prabh *et al.* 2018; Rödelsperger 2021; Wighard *et al.* 2022).

While we saw the greatest variation between *P. pacificus* and *C. elegans* in terms of histone gene count, our results also reveal differences in the types of histone-modifying enzymes present. Most surprisingly, we were unable to identify *P. pacificus* orthologs of the PRC2 components MES-2 and MES-3 despite searching with several independent methods: orthology clustering (OrthoFinder), BLASTP, TBLASTN into the genome, TBLASTN into the transcriptome, and manual inspection of RNA-seq reads mapped to the genomic locus. We did identify a *mes-6* pseudogene in *P. pacificus*, providing further evidence of a nonfunctional *P. pacificus* PRC2 since MES-6, MES-2, and MES-3 are each required for PRC2 catalytic function (Jiao and Liu 2015; Ahringer and Gasser 2018; Snel *et al.* 2022). It is formally possible that *mes-2* and *mes-6* genes duplicated, were integrated into a nonassembled part of the genome, and their respective parental genes pseudogenized. However, we should still be able to recover tBLASTn hits from the raw sequencing reads and could not. Moreover, missing genes should appear randomly across the phylogeny and should also be random regarding the presence/absence of constituent members of complexes. The phylogenetic signature of a single loss and the fact that no intact members of the complex are recoverable within a phylogenetically well-defined cluster of closely related species argue against this possibility. Thus, although we were initially skeptical that the PRC2 complex was absent in *P. pacificus*, we ultimately could not refute this interpretation with multiple independent methods. We conclude that the *mes-2* and *mes-6* members of the PRC2 complex were lost in the last common ancestor of *P. pacificus* and *P. arcanus*. To our knowledge, the complete loss of PRC2 has yet to be documented for any multicellular organism. This raises the question of how the loss of PRC2 is tolerated in *P. pacificus*. The few eukaryotes without PRC2 (the single-celled yeasts *S. pombe* and *S. cerevisiae*) have simultaneously lost H3K27me3 (Shaver *et al.* 2010). However, we confirmed using LC-MS/MS and immunofluorescence that H3K27me3 is present in *P. pacificus*. This finding suggests that an enzyme other than PRC2 must catalyze H3K27me3 in *P. pacificus*.

Evolutionary changes in the composition and functionality of PRC2 complex proteins have been documented between invertebrates and

humans. For example, the main SET methyltransferase protein was duplicated in vertebrate evolution, leading to two genes: EZH1 and EZH2. EZH2 is most similar to invertebrate orthologs and is responsible for the majority of methyltransferase activity in mammals; therefore, we focused our analyses on this gene (Fischer *et al.* 2022). SUZ12/MES-3 appears to be the most evolutionary labile subunit, presumably because of its role primarily as a structural scaffold rather than the enzymatic component (EZH2/MES-3) or the "reader" of H3K27 methylation (EED/MES-6) (Fischer *et al.* 2022). Indeed, until recently MES-3 was considered a *C. elegans*-specific PRC2 subunit and not a true SUZ12 ortholog (Snel *et al.* 2022). While proteins similar to SUZ12 but not MES-3 have been reported in the nematodes *B. malayi* and *T. spiralis,* our orthology analysis did not recover either of these (Snel *et al.* 2022; Supplementary File 4), presumably due to a difference in methods. We were also unable to identify any MES-3/SUZ12 orthologs in any *Pristionchus* nematodes. We envision two scenarios on either end of the evolutionary spectrum to explain this observation: (1) an MES-3/SUZ12 homolog is present in *Pristionchus* species but is too diverged to identify via orthology or (2) MES-3/SUZ12 was lost prior to the divergence of the *Pristionchus* genus. In all cases where it has been investigated, all three components are necessary for methyltransferase activity (Cao *et al.* 2002; Cao and Zhang 2004; Montgomery *et al.* 2005; Højfeldt *et al.* 2018). Thus, the second scenario seems unlikely unless another scaffold protein or HMT took its place. In that case, the reduction in methyltransferase activity could have been compensated for by the uncharacterized H3K27me3 HMT in *Pristionchus*. However, that would not explain the delayed loss of *mes-2* and *mes-6* relative to *mes-3*/SUZ12. Thus, we currently favor a third scenario that is in between these extremes: there was a highly diverged homolog, but it was lost at a similar evolutionary branch to the loss of *mes-2* and *mes-6*.

In *C. elegans*, PRC2 is primarily expressed in the germline, though with some expression recently reported in the endoderm and mesoderm (Bender *et al.* 2004; Engert *et al.* 2018; van der Vaart *et al.* 2020). Knocking out *mes-2* leads to a maternal effect sterile phenotype and reduces H3K27me3 in the germline. However, in these experiments, H3K27me3 is not lost in the soma, indicating that in *C. elegans,* MES-2 is responsible for maintaining germline but not somatic H3K27me3 (Bender *et al.* 2004). Presumably, another *C. elegans* H3K27me3 methyltransferase is responsible for maintaining somatic H3K27me3, though to date, the identity of this somatic writer is unknown. Our immunostaining experiments in *P. pacificus* demonstrate that, despite the absence of PRC2, H3K27me3 is still maintained in both the gonad and somatic tissue. One possibility is that both *P. pacificus* and *C. elegans* share an uncharacterized histone methyltransferase, responsible for somatic H3K27me3 in *C. elegans* and global H3K27me3 in *P. pacificus.* Of course, these functions and their enzymatic writers may be unrelated. Lastly, H3K27me3 is highly enriched over the super-gene locus controlling mouth-form in *P. pacificus,* suggesting a role for H3K27me3 in maintaining proper expression of genes related to mouth-form plasticity (Werner *et al.* 2023).

Going forward, our inventory of putative epigenetic genes will be a valuable resource for identifying the *P. pacificus* "PRC2 equivalent" (particularly those containing SET domains, a typical histone lysine methyltransferase domain). One strategy would be to systematically knock out all SET-containing genes and test for loss of H3K27me3. However, there are more than 40 SET-containing genes, and it may be useful to filter this list for genes with similar structural domains as EZH2 (via structural approaches like AlphaFold) or increased positive selection correlating with the loss of PRC2 in the *Pristionchus* lineage. Though this

uncharacterized methyltransferase is likely conserved in *P. pacificus, P. exspectatus, and P. arcanus* at minimum, another formal possibility is that the "PRC2 equivalent" is one of thousands of documented *P. pacificus* orphan genes (Athanasouli *et al.* 2023). In this scenario, the gene may have "fallen out" of our orthology pipeline unless containing a characteristic HMT domain (e.g. SET). To account for this possibility, an unbiased strategy of biochemical purification of *P. pacificus* proteins with H3K27me3 catalytic or binding properties may be useful. In general, identifying the enzyme responsible for H3K27me3 in *P. pacificus* will expand our understanding of histone-modifying enzymes' evolutionary pressures and catalytic solutions.

## Data availability

The LC-MS/MS has been deposited to the ProteomeXchange Consortium via the PRIDE partner repository with the dataset identifier PXD046748. The code and data used to create the histone position and synteny plots can be found at https://github.com/audreybrown1/Brown-et-al.-2023-library. All other data needed to evaluate the conclusions in this paper are included in Supplementary files, which are referenced throughout the text. Supplementary File 1 lists the reference Pfam domains and Supplementary File 2 contains the reference model-organism epigenetic genes used in the domain and orthology-informed pipeline. Supplementary File 3 contains the InterProScan results for *P. pacificus*. Supplementary File 4 contains the initial OrthoFinder Orthogroup results. Supplementary File 5 contains gene position information for any manually reannotated genes. Supplementary File 6 contains our predicted putative epigenetic genes for *P. pacificus*. Supplementary File 7 contains the InterProScan results for *C. elegans* and Supplementary File 8 contains our predicted putative epigenetic genes for *C. elegans.* Supplementary File 9 contains all references on histone-modifying enzymes used to generate Tables 2 and 3. Supplementary File 10 contains the OrthoFinder Orthogroup results using *Pristionchus* species. Supplementary Files 11 and 12 contain gene position data used to generate the *mes-2* and *mes-6* synteny plots (Fig. 5b and c). Supplementary Files 13 and 14 contain the histone gene position information used to generate the histone position plots (Fig. 2e and f).

Supplemental material is available at GENETICS online.

## Acknowledgments

We thank past and present members of the Werner lab for all input and discussions about this project, especially Madelyn Purnell for the many discussions about gene evolution. We also thank Ralph Sommer for guidance and advice. We would also like to acknowledge WormBase, and the editor and reviewers for their thoughtful commentary.

## Funding

This work was supported by the National Institute of General Medical Sciences (R35GM150720) and startup funds from the School of Biological Sciences at the University of Utah. ALB was also supported by a National Institutes of Health funded training grant (T32-GM122740) and the National Science Foundation Graduate Research Fellowship (DGE-2039655). CJW received support from the Royal Society Dorothy Hodgkin Fellowship. OR is supported by a National Institute of General Medical Sciences

grant R35GM128804. We also acknowledge the Max Planck Institute for funding the mass spectrometry.

## Conflicts of interest

The author(s) declare no conflicts of interest.

## Author contributions

ALB, ABM, CJW, and MSW designed the bioinformatic analysis pipeline; ALB and ABM performed bioinformatic identification of epigenetic genes; ALB and MW analyzed histone gene cluster conservation; ALB performed all analyses on PRC2 conservation; MF-W performed LC-MS/MS with guidance from BM. ALB created all phylogenies with guidance from CJW. ALB and SG performed immunofluorescence experiments with guidance from OR. Writing was by ALB and MSW with input from all authors.

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

*Editor: V. Reinke*