## [Peer Review File · Genetics]

Characterization of the *Pristionchus pacificus* "epigenetic toolkit" reveals the evolutionary loss of the histone methyltransferase complex PRC2

Audrey Brown, Adriaan Meiborg, Mirita Franz-Wachtel, Boris Macek, Spencer Gordon, Ofer Rog, Cameron Weadick, and Michael Werner

NOTE: The reviews and decision letters are unedited and appear as submitted by the reviewers.

In extremely rare instances and as determined by a Senior Editor or the EIC, portions of a review may be redacted. If a review is signed, the reviewer has agreed to no longer remain anonymous.

The review history appears in chronological order.

Review Timeline:

Submission Date:	2023-12-04
Editorial Decision:	2024-01-16
Resubmission Received:	2024-03-01
Accepted:	2024-03-05

January 15, 2024

GENETICS-2023-306688

Characterization of the *Pristionchus pacificus* "epigenetic toolkit" reveals the evolutionary loss of the histone methyltransferase complex PRC2

Dear Dr. Werner:

Two experts in the field have reviewed your manuscript, and I have read it as well. Overall, the work is considered of broad interest and the finding that H3K27me3 is still present in the absence of PRC2 is very novel. While your manuscript is not currently acceptable for publication in GENETICS, we would welcome a substantially revised manuscript. Both reviewers have comments and concerns to be addressed in a revised manuscript. You can read their reviews at the end of this email. In particular, as emphasized by reviewer 3, the rationale why certain epigenetic proteins were included or excluded needs to be made clearer throughout the manuscript, and the supplemental tables and files that organize these data need to be more comprehensive. We look forward to receiving your revised manuscript. Please let the editorial office know approximately how long you expect to need for revisions.

Upon resubmission, please include:

1. A clean version of your manuscript;
2. A marked version of your manuscript in which you highlight significant revisions carried out in response to the major points raised by the editor/reviewers (track changes is acceptable if preferred);
3. A detailed, point-by-point response to the editor's/reviewers' feedback and to the concerns listed above. Please reference line numbers in this response to aid the editor and reviewers.

Your paper will likely be sent back out for review.

Additionally, please ensure that your resubmission is formatted for GENETICS
<https://academic.oup.com/genetics/pages/general-instructions>

Follow this link to submit the revised manuscript: Link Not Available

Sincerely,

Valerie Reinke
Associate Editor
GENETICS

Approved by:
Audrey Gasch
Senior Editor
GENETICS

Reviewer #1 (Comments for the Authors (Required)):

Brown et al. identified and curated the "epigenetic toolkit" for nematode *P. pacificus*. They presented a thorough study of an interesting case of gene loss/pseudogenization in the nematode *P. pacificus*. Curiously, after rigorous search, the authors found that the histone methyltransferase complex PRC2 was lost in *P. pacificus* its sister *Pristionchus* species. The three genes within the complex were likely lost/pseudogenized through different evolutionary mechanisms. Yet, the product of PRC2, H3K27me3 is preserved in *P. pacificus*, suggesting that "the players may have changed but the game remains the same" and that an unknown methyltransferase replaced the role of PRC2 in *P. pacificus*.

I only have minor suggestions.

Page 25 line 535, "No mes-3 orthologs could be found in any of the diplogastrid examined by any method, indicating that this gene is either absent, or so diverged from the human and *C. elegans* sequences". Please specify the thresholds used to identify orthologs. Have the authors tried relaxing the ortholog searching criteria for mes-3?

Figure 4C caption says MES-3/SUZ12 orthogroups, but in the figure only SUZ12 genes were plotted.

Figure 4E, I am curious about the evolutionary story of PRC2 in nematodes. It seems that *mes-2* and *mes-6* are experiencing ongoing losses, as some *Pristionchus* species still retain a functional copy. Yet *mes-3* is nowhere to be found in the *Pristionchus* lineage, and *C. elegans mes-3* is already highly diverged from SUZ12. Do other nematodes contain a copy of *mes-3*? Maybe the authors could discuss a little more on why the three genes in the complex were lost at different rate.

Reviewer #2 (Comments for the Authors (Required)):

The manuscript by Brown et al characterizes the epigenetic toolkit of the nematode model organism *Pristionchus pacificus*. This nematode has become a very important model system for the study of developmental plasticity given a mouth form polyphenism that results in two alternative mouth morphs. As *C. elegans* does not have a similar structure, *P. pacificus* and its hermaphroditic mode of reproduction represent a unique system to study the molecular biology associated with phenotypic plasticity. This manuscript provides a comprehensive analysis of the epigenetic toolkit of this worm, which will be extremely useful for all future studies of mouth form plasticity. Therefore, this paper be a very valuable resource and will receive many citations.

The authors have done a careful bioinformatic analysis that is placed in a proper comparative and phylogenetic context given the many genomes available in the genus *Pristionchus*. Most excitingly, this study shows the absence of the PRC2 histone methyltransferases, *mes-2* and *mes-6*, which have been lost in *P. pacificus* and some of its close relatives, but not in the complete genus. Interestingly, both genes were lost by different mechanisms. However, biochemical analysis revealed that the H3K27me3 mark is still present indicating that another gene must encode for an enzyme writing this histone mark. I am not aware of any other multicellular organism, in which this complex has been secondarily lost.

The combination of this useful inventory of the epigenetic toolkit and the extremely unexpected finding of the PRC2 complex loss, make this manuscript very attractive. However, I find the Discussion in its current form not to be fully developed. I support publication, in principle, after several issues have been properly addressed in the Discussion:

1. The authors describe a DNMT gene in *P. pacificus*, but do not discuss DNA methylation.
2. I would want to see a discussion on how the authors will search for the PRC2 complex 'equivalent'. Will they systematically knockout all HMTs? Are there other systematic approaches that might help identifying the corresponding gene/enzyme. This should not be a detailed description of their next steps in the research program, but a general brainstorming would indicate to the reader that important follow up steps will be highly rewarding.
3. In this context it would also be good to return to the question of the quality of the bioinformatic pipeline: Can the authors be sure that the PRC2 complex 'equivalent' will be among the identified proteins, or could it be that the corresponding gene has been 'lost' in their workflow (Fig. 1)?
4. Also in this context: What about the many thousands of orphan genes in *Pristionchus*? Could they harbor the responsible activity? Could AlfaFold be of any help in this context?

It would help the reader to obtain a more sophisticated picture on what the authors think and how they want to approach this very interesting puzzle. That is what Discussion sections are good for.

Reviewer #3 (Comments for the Authors (Required)):

This paper annotates some classes of epigenetic proteins of *Pristionchus pacificus*, among which are histone methyltransferases, and they find that *P. pacificus* lacks proteins of the PRC2 complex, including the EZH2 enzyme that catalyses H3K27me3. Despite this, the authors show that *P. pacificus* has detectable H3K27me3, indicating that an unknown enzyme has this function.

There is some value in the identification of some classes of *P. pacificus* epigenetic proteins, but it is not appropriate to call the work a characterization the *Pristionchus pacificus* "epigenetic toolkit." As the paper progresses and through careful examination of the tables, it becomes clear that the paper focuses on only the subset of proteins that affect the covalent modification of histones by methylation, acetylation, phosphorylation, or ubiquitylation, the covalent modification of DNA, or RNAi. It is unclear why the authors have not included proteins other classes of epigenetic proteins such as proteins that bind histone modifications, nucleosome remodellers (i.e., proteins that insert, move, or evict nucleosomes), or most components of histone modification complexes (e.g., COMPASS, PRC1, PRC2) in their computational analyses. Though ideally the analysis would include a broader cross section of proteins, at a minimum, the narrow focus must be made clear in the text from the outset, and the rationale for focusing on this particular subset of epigenetic proteins must be explained. The main text describing the analysis of epigenetic proteins lacks specific details and citation of tables, making it difficult for the reader to follow exactly what has been done.

Through their analyses, they discovered that *P. pacificus* lacks an EZH2 ortholog, which is the only known eukaryotic enzyme to catalyze H3K27me3. They then look for other PRC2 components and find those lacking as well, indicating PRC2 is not present, and through phylogenetic analyses, they show when the loss occurred. Using antibodies and mass spec analyses, they show that despite the loss of PRC2, *P. pacificus* does contain H3K27me3, indicating an unknown enzyme likely carries out this modification. This finding will be of interest to those studying this modification.

Specific comments:

Line 34: Where can the reader find the list of 260 orthogroups with reference model species epigenetic proteins?

It is interesting that only ~50% of orthogroups contain a *P. pacificus* protein (and as similar fraction was seen for *C. elegans*). Please comment on which orthogroups/gene types are not present.

Line 125: Please refer to the file containing the reference epigenetic-associated protein domain dataset used.

Line 222: Please add the concentration of Arg-C used.

Line 228: I don't understand what "Endoprotease ArgC" means here. Is it fragments cleaved by ArgC? Please clarify.

Line 243: Suggest changing title of this section as it discusses "writers" and "erasers"

Line 248: What is meant by "For brevity"?

Line 309: Clarify the fixation protocol. Were embryos fixed while compressed under a coverslip?

Lines 336 - 339: Please add citation to the relevant file.

Line 342: It is surprising that only half of the 260 orthogroups of model species epigenetic proteins contained a *P. pacificus* protein. It would be of interest for the authors to comment on the groups or types of groups with and without a *P. pacificus* protein, and similarly to comment on this feature for *C. elegans* as well, including whether the similarity between the two species.

Line 361: Which 105 *C. elegans* putative epigenetic readers were missing from the original *C. elegans* reference set? Please refer to the file where this is annotated. Is there a simple reason they were not initially included?

Line 365: Which 58 *C. elegans* proteins had no explicit epigenetic annotations and why are they likely the result duplication and divergence? Please refer to the file where this is annotated.

Line 367: Please strengthen the evidence for the claim that "our pipeline is more effective at identifying epigenetic proteins in well-annotated organisms than literature or database searches and 2) our pipeline can identify "new" putative epigenetic proteins, even in organisms with decades of functional characterization" or clarify the statements. Isn't the "pipeline" a collection of database searches? Which "new" epigenetic proteins were identified? The claims seem overstated.

Line 375: Please highlight some examples of instances of proteins that have no ortholog within the other species. Are these where there is no 1:1 ortholog or where the gene type is not present and could not have been found by homology to a protein that is similar but not orthologous? It is not clear how Supp Fig. 1 confirms that inclusion of species beyond *C. elegans* was necessary to identify proteins without orthologs.

Table 1: Please point out the obvious. First, the two species harbor at least one member of each family, with the exception of DNMT for *C. elegans*. Second, PRMT genes have the largest difference in gene number. I do not understand why statistics are being used to make comparisons. Why is it of interest that gene number changes are statistically significant? Aren't any changes of interest?

Supplementary Figure 4 needs a legend.

Are the authors able to provide the abundance of each modification from the MS data?

Table 2 shows some oddities: none of the *C. elegans* histone deacetylase genes are listed (*hda-1* to *hda-6*, *hda-10*, and *hda-11*); *met-2/SETDB1* is listed twice. the two entries have different *P. pacificus* genes listed and the second entry has no histone modification listed. There may be other issues. Please check the table carefully.

Line 483 The first parenthesis should be moved so that the parentheses just encircle the references.

Line 488: Add reference to Gaydos and Strome, 2014 and to Kaneshiro et al 2022, which showed that H3K27me3 is

transgenerationally inherited in *C. elegans*.

line 499: Does homology of most ortholog pairs extend well outside the set domain?

Figure 4E: There is a typo in the first column of Figure 4E, as it is labelled EZH2/MES-6.

Line 586: I didn't understand the point the authors were trying to make regarding the punctate nature of the H3K27me3 staining. What was meant by "punctate"? Did they mean that H3K27me3 did not cover all chromatin marked by H3? This is known to be the case in *C. elegans*.

Please add names at the top of all additional files to help the reader, and if possible include a list of additional files and their descriptions somewhere in the paper

Please provide legends for every table (e.g., on a separate tab), and spell out all abbreviations (e.g., HMT etc).

Associate Editor Comments:

Reviewer #1 (Comments for the Authors (Required)):

Brown et al. identified and curated the "epigenetic toolkit" for nematode *P. pacificus*. They presented a thorough study of an interesting case of gene loss/pseudogenization in the nematode *P. pacificus*. Curiously, after rigorous search, the authors found that the histone methyltransferase complex PRC2 was lost in *P. pacificus* its sister *Pristionchus* species. The three genes within the complex were likely lost/pseudogenized through different evolutionary mechanisms. Yet, the product of PRC2, H3K27me3 is preserved in *P. pacificus*, suggesting that "the players may have changed but the game remains the same" and that an unknown methyltransferase replaced the role of PRC2 in *P. pacificus*.

We thank the reviewer for the positive evaluation of our manuscript, and provide detailed answers to their suggestions below.

I only have minor suggestions.

Page 25 line 535, "No *mes-3* orthologs could be found in any of the diplogastrid examined by any method, indicating that this gene is either absent, or so diverged from the human and *C. elegans* sequences". Please specify the thresholds used to identify orthologs. Have the authors tried relaxing the ortholog searching criteria for *mes-3*?

We thank the reviewer for the suggestion. We tried relaxing our BLAST threshold (for all BLASTP and TBLASTN based analyses) to $10e-1$ while searching for *mes-3*. Even with this relaxed threshold, we found no hits using either the human *SUZ12* or *C. elegans mes-3/SUZ12* sequences as query. We think this is valuable information for the reader, and indeed it has come up in Q & A sessions when giving talks in our department, and thus have added descriptions of this analysis to the Results on lines 602-605, and Methods on lines 296-299. *Mes-3* is so diverged from *SUZ12* that it was only recently appreciated (Snel et al., iScience. 2022) that it was indeed a homolog so we were not entirely surprised to not observe it in *Pristionchus*. The absence of *mes-2* and *mes-6* on the other hand (which do have clear 1:1 orthologs in *C. elegans*) was much harder to believe. Because *SUZ12*, at least in vertebrates, appears to be a scaffold for the complex, rather than the enzymatic component (EZH1/2 or MES-2) or the 'reader' of H3K27me2/3 (EED or MES-6), one could imagine that it is under less evolutionary constraint than the other two.

Figure 4C caption says MES-3/SUZ12 orthogroups, but in the figure only SUZ12 genes were plotted.

We thank the reviewer for pointing this out. We have updated the caption to read only "SUZ12" and added text in the figure caption to emphasize that the *C. elegans* MES-3 formed an isolated orthogroup containing only itself (hence why it is not plotted).

Figure 4E, I am curious about the evolutionary story of PRC2 in nematodes. It seems that *mes-2* and *mes-6* are experiencing ongoing losses, as some *Pristionchus* species still retain a functional copy. Yet *mes-3* is nowhere to be found in the *Pristionchus* lineage, and *C. elegans mes-3* is already highly diverged from *SUZ12*. Do other nematodes contain a copy of *mes-3*? Maybe the authors could discuss a little more on why the three genes in the complex were lost at different rate.

We agree that there is something interesting going on with PRC2 in nematodes, especially the *SUZ12/MES-3* unit. To address this, we have added new text to the Discussion starting on line 716 detailing what is currently known about *SUZ12/MES-3* evolution, and speculating on why these genes may have been lost at different rates (see above response to first suggestion). Specifically, we point out

that SUZ-12/MES-3 seems to be the most evolutionary variable subunit of PRC2, presumably because of its role as a structural scaffold in the complex in contrast to the enzymatic or reader components. In our hands, SUZ12 orthologs found in various Clade III/IV nematodes display greater sequence-level conservation than MES-3 orthologs from *Caenorhabditis*. This implies that something happened to MES-3/SUZ-12 specifically in Rhabditina lineages. It is possible that there is a *mes-3* homolog in *P. pacificus* or other *Pristionchus* nematodes that we were unable to detect, or perhaps more likely, an equally diverged homolog of SUZ12 that will take additional measures (perhaps through structural homology) to identify. Another possibility is that a SUZ12 homolog was lost earlier, and that retention of HMT activity in *Pristionchus* was compensated by a combination of residual *mes-2* and *mes-6* activity and members of the currently unknown H3K27 methyltransferase complex in *Pristionchus*. However, we believe that the most likely scenario is that there was a highly diverged SUZ12 homolog which was recently lost in parallel with *mes-2* and *mes-6* in *P. pacificus*. In this event, that homolog should still be present in more basal *Pristionchus* species, and is an issue we hope to investigate in depth, in the future.

Reviewer #2 (Comments for the Authors (Required)):

The manuscript by Brown et al characterizes the epigenetic toolkit of the nematode model organism *Pristionchus pacificus*. This nematode has become a very important model system for the study of developmental plasticity given a mouth form polyphenism that results in two alternative mouth morphs. As *C. elegans* does not have a similar structure, *P. pacificus* and its hermaphroditic mode of reproduction represent a unique system to study the molecular biology associated with phenotypic plasticity. This manuscript provides a comprehensive analysis of the epigenetic toolkit of this worm, which will be extremely useful for all future studies of mouth form plasticity. Therefore, this paper be a very valuable resource and will receive many citations.

The authors have done a careful bioinformatic analysis that is placed in a proper comparative and phylogenetic context given the many genomes available in the genus *Pristionchus*. Most excitingly, this study shows the absence of the PRC2 histone methyltransferases, *mes-2* and *mes-6*, which have been lost in *P. pacificus* and some of its close relatives, but not in the complete genus. Interestingly, both genes were lost by different mechanisms. However, biochemical analysis revealed that the H3K27me3 mark is still present indicating that another gene must encode for an enzyme writing this histone mark. I am not aware of any other multicellular organism, in which this complex has been secondarily lost.

The combination of this useful inventory of the epigenetic toolkit and the extremely unexpected finding of the PRC2 complex loss, make this manuscript very attractive. However, I find the Discussion in its current form not to be fully developed. I support publication, in principle, after several issues have been properly addressed in the Discussion:

We thank the reviewer for their positive evaluation and for their suggestions to improve the manuscript and Discussion. We believe the overall manuscript is now stronger, and provide detailed responses to each comment below:

1. The authors describe a DNMT gene in *P. pacificus*, but do not discuss DNA methylation.

Indeed, and the *P. pacificus* DNMT gene did catch our attention. The presence of this gene in *P. pacificus* has actually been identified by an earlier study (Gutierrez and Sommer 2004), hence why we initially chose not to focus on it in this manuscript. However, we have now included text in the Results describing this finding (beginning on line 441) and briefly mention the results from Gutierrez and Sommer 2004. In short, there is as yet no evidence that *P. pacificus* exhibits 5mC methylation. Therefore, what function it has, if any, remains unknown.

2. I would want to see a discussion on how the authors will search for the PRC2 complex 'equivalent'. Will they systematically knockout all HMTs? Are there other systematic approaches that might help identifying the corresponding gene/enzyme. This should not be a detailed description of their next steps in the research program, but a general brainstorming would indicate to the reader that important follow up steps will be highly rewarding.

We have added a new paragraph at the end of the Discussion addressing this question, beginning on line 750, and agree that it is important to convey to the reader a sense that "more is to come."

3. In this context it would also be good to return to the question of the quality of the bioinformatic

pipeline: Can the authors be sure that the PRC2 complex 'equivalent' will be among the identified proteins, or could it be that the corresponding gene has been 'lost' in their workflow (Fig. 1)?

We agree that this is an important caveat to be aware of going forward, and have addressed this concern in the new Discussion paragraph where we discuss future directions (line 759). In essence, we think it probable that the PRC2 “equivalent” is in our identified “toolkit”. One data point to support this is the expansive number of genes we recovered from *C. elegans* using our pipeline compared to previous: we identified 265 putative *C. elegans* epigenetic genes compared to 160 reference *C. elegans* epigenetic genes. A second is that the enzymatic component likely contains a SET domain, which have highly conserved catalytic residues, and therefore our InterProScan analysis should have identified it. Nevertheless, in the case that the PRC2 equivalent has ‘slipped through the cracks’, a biochemical approach should be an unbiased way to identify the PRC2 equivalent (i.e., pull-down with H3K27me2/3 peptide or nucleosome, or isolating proteins by fractionation that have H3K27me2/3 HMT activity).

4. Also in this context: What about the many thousands of orphan genes in *Pristionchus*? Could they harbor the responsible activity? Could AlfaFold be of any help in this context?

Indeed, this is also a possibility. Because PRC2 has been lost in the sister species *P. arcanus* and *P. expectatus*, we suspect that the new HMT protein is likely conserved in these other two species at least. In this vein, another potential avenue would be to identify candidate genes with an evolutionary signature (such as increased positive selection) correlating with the loss of PRC2, but agnostic to homology to other methyltransferases. If an orphan *P. pacificus* gene is responsible for K27 methylation, it may not harbor orthology to previously characterized HMT genes, and could have been missed in our bioinformatic analysis (as mentioned above). However, if it contained a characteristic HMT domain (e.g., SET), then it still would have been picked up from the ‘epigenetic domain’ part of the pipeline.

Alpha Fold and comparative structure approaches (i.e., Dali, Foldseek, SWISS-MODEL, HOMELETTE) may be useful tools for analyzing both orphan and non-orphan genes: if any bear a similar catalytic or histone binding structure to PRC2 components, they are potentially good candidates to follow up on with knockout experiments. We have included these points in the last Discussion paragraph beginning on line 750.

It would help the reader to obtain a more sophisticated picture on what the authors think and how they want to approach this very interesting puzzle. That is what Discussion sections are good for.

We thank the reviewer and believe that these suggestions have strengthened our Discussion section, and the manuscript writ large.

Reviewer #3 (Comments for the Authors (Required)):

This paper annotates some classes of epigenetic proteins of *Pristionchus pacificus*, among which are histone methyltransferases, and they find that *P. pacificus* lacks proteins of the PRC2 complex, including the EZH2 enzyme that catalyzes H3K27me3. Despite this, the authors show that *P. pacificus* has detectable H3K27me3, indicating that an unknown enzyme has this function.

There is some value in the identification of some classes of *P. pacificus* epigenetic proteins, but it is not appropriate to call the work a characterization the *Pristionchus pacificus* "epigenetic toolkit."

As the paper progresses and through careful examination of the tables, it becomes clear that the paper focuses on only the subset of proteins that affect the covalent modification of histones by methylation, acetylation, phosphorylation, or ubiquitylation, the covalent modification of DNA, or RNAi. It is unclear why the authors have not included proteins other classes of epigenetic proteins such as proteins that bind histone modifications, nucleosome remodelers (i.e., proteins that insert, move, or evict nucleosomes), or most components of histone modification complexes (e.g., COMPASS, PRC1, PRC2) in their computational analyses. Though ideally the analysis would include a broader cross section of proteins, at a minimum, the narrow focus must be made clear in the text from the outset, and the rationale for focusing on this particular subset of epigenetic proteins must be explained.

We thank the reviewer and believe that their suggestions have strengthened our study. We do appreciate that the current scope omits proteins that act upon epigenetic information carriers to carry out their function. Indeed, we went back and forth on this issue several times. We believe the strategy that we settled on enabled us to evaluate how epigenetic machinery evolves and identify prime candidates for further analysis of epigenetic processes - without creating lists of thousands of proteins which would effectively dilute the focus of our study. **Nevertheless, we have added an analysis of the enzymes that perform nucleosome remodeling in the revision (line 367-369, Table 1, Additional File 6).** We hope that this strikes a middle ground which is more inclusive but also consistent with our prior approach. Below, we attempt to justify our rationale for focusing on the enzymatic writers and erasers, and now remodelers. **We have also added more text on lines 367-374 in the results and line 51 in the introduction to emphasize our rationale for this focus.**

Principally, we used the modern (and admittedly somewhat narrow) definition of epigenetics as "mitotically and/or meiotically stable, non-DNA sequence based mechanisms that regulate gene expression" (Deans & Maggert, Genetics, 2004) for the foundation of our study (line 49-50). Thus, we focused on the three accepted mechanisms of epigenetic information transfer: DNA methylation, histone modifications and small RNAs. With the focus on epigenetic information carriers, we opted to identify histones (the canvas), the enzymes that 'write' or 'erase' histone or DNA modifications or are involved in small RNA biogenesis, and (now added in the revision) nucleosome remodelers. By focusing on enzymatic writers and erasers we could make straightforward comparisons of the presence or absence of epigenetic marks, and then investigate functionally related proteins in a case-by-case basis. For example, this strategy identified the absence of the H3K27me2/3 writer *ezh2/mes-2*, which led us to the discovery that the entire PRC2 complex was lost in *P. pacificus*.

A second rationale for focusing on *enzymatic* writers, erasers and small RNA biogenesis is the issue of deciding which genes should be considered to have epigenetic function. For example, in the second half of the manuscript we focus on the core PRC2 complex which contains three components necessary for methylation *in vitro*; but *in vivo* the complex contains at minimum seven stable members,

plus transient interactions with several more proteins (Margueron and Reinberg, 2013). Several of these proteins are also members of other complexes which have little or nothing to do with K27 methylation. Additionally, many reader-protein domains that bind to modified histones can also bind to modifications on non-histone proteins (Barman et al., JBC. 2024). Ultimately, we realized that attempting to draw clean lines between epigenetic and non-epigenetic function was a slippery slope. By cataloging all proteins that have a potential role in epigenetic mechanisms – some tangentially and some probably not at all - would be casting such a wide net as to dilute the most relevant orthologs in our study. On the other hand, by identifying the enzymes behind epigenetic information transfer (i.e., the SET-containing catalytic component of COMPASS), we can hypothesize that the critical members of those complexes, as well as the functional output (i.e., H3K4 methylation) exist in *P. pacificus*.

Nevertheless, again, we do appreciate the reviewers point and have added the enzymatic components of nucleosome remodelers to our study. This resulted in the addition of 8 new genes to both our *P. pacificus* and *C. elegans* datasets. We have correspondingly updated all tables, statistics, additional files, and numbers reported in the main text. We hope that these revisions, and the rationale provided above (and in the updated main text: **lines 367-374 and 51**) make the case that this study represents the epigenetic ‘toolkit’ of *P. pacificus* (but not the epigenetic ‘tool shed’).

The main text describing the analysis of epigenetic proteins lacks specific details and citation of tables, making it difficult for the reader to follow exactly what has been done.

We have updated our manuscript in several places to address this point (lines 132, 138, 158-165, 232-233, 275, 301, 321, 377, 381, 390, 392, 404, 410, 412, and 415), and have addressed all specific comments regarding the citation of tables.

Through their analyses, they discovered that *P. pacificus* lacks an EZH2 ortholog, which is the only known eukaryotic enzyme to catalyze H3K27me3. They then look for other PRC2 components and find those lacking as well, indicating PRC2 is not present, and through phylogenetic analyses, they show when the loss occurred. Using antibodies and mass spec analyses, they show that despite the loss of PRC2, *P. pacificus* does contain H3K27me3, indicating an unknown enzyme likely carries out this modification. This finding will be of interest to those studying this modification.

Specific comments:

Line 34: Where can the reader find the list of 260 orthogroups with reference model species epigenetic proteins?

We thank the reviewer for pointing this out. We have added this information to Additional File 4, and referenced this file at the appropriate places in the main text (line 138 and line 390).

It is interesting that only ~50% of orthogroups contain a *P. pacificus* protein (and as similar fraction was seen for *C. elegans*). Please comment on which orthogroups/gene types are not present.

This trend seems to be a general feature across all orthogroups (Additional File 4), not just those containing *P. pacificus* or *C. elegans*, or those containing epigenetic proteins. Below we summarize some orthogroup statistics:

Subset	Number (%)
All orthogroups	52521 (100%)
Fully inclusive orthogroups (contain all species)	913 (1.7%)
Mostly inclusive orthogroups (contain >50% of species)	4609 (8.7%)
Species-specific orthogroups (contain just one species)	24187 (46%)
P. pacificus specific orthogroups	409 (0.7%)
C. elegans specific orthogroups	1621 (3.1%)

In summary, we find that the majority of orthogroups did not contain representatives from every single species. Furthermore, nearly half of the total orthogroups (46%) were species-specific, containing representatives from only one species. Presumably, these represent clade-specific orthogroups where no gene orthologs are present in any of the other included species. This appears to be the major reason why only ~50% of our 261 (updated number) epigenetic-containing orthogroups have a *P. pacificus* gene. **Of these 261 orthogroups, 67.6% contained genes from only one or two species.** Our conclusion is that these orthogroups are clade-specific. We did not observe any obvious trends in the types of functional gene families present or absent in the epigenetic orthogroups lacking *P. pacificus* or *C. elegans* genes. This information has been added to the results section starting on line 385.

Line 125: Please refer to the file containing the reference epigenetic-associated protein domain dataset used.

We have made the appropriate change (now on line 132).

Line 222: Please add the concentration of Arg-C used.

We have added this information to the Methods section, as well as additional text clarifying the protocol (line 241).

Line 228: I don't understand what "Endoprotease ArgC" means here. Is it fragments cleaved by ArgC? Please clarify.

This is correct, we have now clarified the text to read "For the detection of peptides cleaved by Arg-C, we selected proteolytic fragments with a maximum of two and three missed cleavages." (line 251)

Line 243: Suggest changing title of this section as it discusses "writers" and "erasers"

We have made the appropriate change (now at line 266).

Line 248: What is meant by "For brevity"?

Because histone acetylation and methylation are the most widely studied histone PTMs, we chose to focus on this group. This has been clarified in the text line on line 271.

Line 309: Clarify the fixation protocol. Were embryos fixed while compressed under a coverslip?

We have clarified the protocol starting on line 340. In summary, worms were picked into a drop of Egg buffer on a coverslip, dissected, and then fixed by adding another drop of Egg buffer + 1% formaldehyde. Then the coverslip containing the now-fixed worms was transferred onto HistoBond slides.

Lines 336 - 339: Please add citation to the relevant file.

We have added the appropriate citation, now on line 381.

Line 342: It is surprising that only half of the 260 orthogroups of model species epigenetic proteins contained a *P. pacificus* protein. It would be of interest for the authors to comment on the groups or types of groups with and without a *P. pacificus* protein, and similarly to comment on this feature for *C. elegans* as well, including whether the similarity between the two species.

We believe that our response to the reviewer's previous comment regarding ~50% of orthogroups containing *P. pacificus* proteins addresses this question. As stated above (and indicated in the revised text), this pattern appears to be due less to the inclusion or exclusion of specific classes of genes and is a general feature of many orthogroups being species-specific or clade specific.

Line 361: Which 105 *C. elegans* putative epigenetic readers were missing from the original *C. elegans* reference set? Please refer to the file where this is annotated. Is there a simple reason they were not initially included?

We have added this information to Additional File 8, and referenced it at the appropriate place in the text on line 415.

When we assembled our reference files, we searched UniProt for entries containing epigenetic annotations, which were found by searching for phrases such as "histone acetyltransferase activity", "DNA demethylase", "RNA-mediated gene silencing", etc. Of the 109 *C. elegans* proteins (updated number that reflects typo corrections) missing from our original reference set, we went back and found that 62 had "epigenetic" annotations on UniProt. Therefore, these represent proteins that we unintentionally overlooked while searching UniProt to assemble our reference file. The other 47 had no explicit epigenetic annotations, hence why these were not initially included. This is why we claim that these may be "new" epigenetic proteins because they had no previous epigenetic annotation. We have updated the text to make the explanation of the initial exclusion of these proteins clearer beginning on line 415, and at line 375.

Line 365: Which 58 *C. elegans* proteins had no explicit epigenetic annotations and why are they likely the result duplication and divergence? Please refer to the file where this is annotated.

We have now annotated these 47 genes (updated number) in Additional File 8, and the reference is given in the text on line 415. Functional protein annotations in UniProt are based on either 1) published studies/experimental data, or 2) similarity with other characterized proteins. Therefore, we conclude that these "epigenetic" proteins with no explicit epigenetic annotations may be cases where epigenetic genes have duplicated and then diverged in a way where they do not bear obvious similarity to other characterized epigenetic genes (and, presumably, UniProt's metrics for determining orthology differ from those in our pipeline). We have clarified this point in the text, and included an example, by writing: "The remaining 47 had no explicit epigenetic annotations and may represent the result of duplication and divergence of other epigenetic-associated genes, or genes where epigenetic function has not been

explicitly experimentally characterized. As an example, we found that in UniProt, the *C. elegans* gene ZK1098.11 is named N-acetyltransferase domain-containing protein 1 (NATD1), presumably because it is an ortholog of a human protein with the same name (Additional File 4). Our analysis revealed that both ZK1098.11 and human NATD1 are orthologs of *A. thaliana* protein At1g77540 (Additional File 4), which possesses in vitro acetyltransferase activity towards both H3 and H4 (Tyler et al. 2006)."

Line 367: Please strengthen the evidence for the claim that "our pipeline is more effective at identifying epigenetic proteins in well-annotated organisms than literature or database searches and 2) our pipeline can identify "new" putative epigenetic proteins, even in organisms with decades of functional characterization" or clarify the statements. Isn't the "pipeline" a collection of database searches? Which "new" epigenetic proteins were identified? The claims seem overstated.

Our pipeline refers to our domain (InterProScan), and orthology (OrthoFinder) methodology as outlined in Figure 1. It is a combination of database searches (InterProScan), orthology prediction (OrthoFinder), and phylogenetic analysis of predicted orthologs. We made these claims while referring to the *C. elegans* genes we identified with no previous epigenetic annotations in UniProt. We have attempted to clarify the text in lines 425-428. It now reads, "Taken together, these results suggest that combining protein domain queries with orthology can better identify epigenetic functions in proteins than either method alone, leading to the recovery of more genes than those currently annotated in databases such as UniProt (even in well-characterized organisms such as *C. elegans*)."

Line 375: Please highlight some examples of instances of proteins that have no ortholog within the other species. Are these where there is no 1:1 ortholog or where the gene type is not present and could not have been found by homology to a protein that is similar but not orthologous? It is not clear how Supp Fig. 1 confirms that inclusion of species beyond *C. elegans* was necessary to identify proteins without orthologs.

We think that the suggestion to include an example here is a good idea and have updated lines 431-439 to add an example and clarify our reference to Supplemental Fig. 1. The new text reads: "Phylogenetic analysis of these families shows that within each, there are several instances of one-to-one orthology between *P. pacificus* and *C. elegans* (Figure S1). However, we also observed multiple instances where this is not the case. For example, the SET containing *P. pacificus* proteins Contig47-snapTAU.201, Iso_D.594.1, PPA40945, and ppa_stranded_DN29031_c0_g1_i1 form a *P. pacificus*-specific clade when visualized in a phylogeny of *P. pacificus* and *C. elegans* SET proteins (Figure S1). While these proteins have no apparent *C. elegans* ortholog, they are members of an orthogroup containing a reference *A. thaliana* protein SUVH10 (Additional File 4) – highlighting the utility of including species beyond *C. elegans* to identify epigenetic proteins in *P. pacificus*."

Table 1: Please point out the obvious. First, the two species harbor at least one member of each family, with the exception of DNMT for *C. elegans*.

Reviewer 2 also suggested including a discussion on the *P. pacificus* DNMT gene. In brief, the presence of a *P. pacificus* DNMT gene has been explored by an earlier study (Gutierrez and Sommer 2004), which is why we chose not to focus on it here. However, we have now included a brief discussion on this point and have also mentioned the difference in *P. pacificus* and *C. elegans* PRMT genes (beginning line 441).

Second, PRMT genes have the largest difference in gene number. I do not understand why statistics are being used to make comparisons. Why is it of interest that gene number changes are statistically significant? Aren't any changes of interest?

We thank the reviewer for this feedback. Our inclusion of statistics was meant to test whether there were any differences in gene numbers across a whole gene family. Such a change would represent a potentially large and deep evolutionary divergence - which ended up being the case with the histone genes. However, we agree that the loss of individual genes is also interesting. We have pared down our use of statistics in the Results and only report the result of the histone 2x2 Fisher's exact test (line 450-454), and instead retain a full description of our statistics in the Methods (line 195). While we agree that we do not need to rely on statistical test results to justify a deep exploration of the histone family — these families are inherently interesting, and it is correct that any changes can be functionally/evolutionarily important — in our experience some readers may take exception to decisions to focus on subsets of the data without statistical backing. In that sense, the inclusion of the statistical test results is an attempt to engage with that group of readers.

Supplementary Figure 4 needs a legend.

We thank the reviewer for highlighting this and have included a legend in the revision.

Are the authors able to provide the abundance of each modification from the MS data?

All raw LC/MS-MS data has been made publicly available on ProteomeXchange under the accession number PXD046748. Unfortunately, we are unable to provide an accurate estimate of the abundance of each modification from this data. Our study summarizes results from several different types of LC/MS-MS processing with different settings. We could potentially compare peptides carrying the same modifications in the same processing. However, many histone peptides are multiply modified. Thus, these measurements would lack abundance information of the same modification on peptides that also bear other modifications. In essence, we could present a table of relative abundances, but it is not a table that we would feel comfortable standing behind.

Table 2 shows some oddities: none of the *C. elegans* histone deacetylase genes are listed (hda-1 to hda-6, hda-10, and hda-11); met-2/SETDB1 is listed twice. the two entries have different *P. pacificus* genes listed and the second entry has no histone modification listed. There may be other issues. Please check the table carefully.

We thank the reviewer for carefully examining our tables and catching these typos. We have made the appropriate corrections.

Line 483 The first parenthesis should be moved so that the parentheses just encircle the references.

We have made the appropriate correction and thank the reviewer for pointing it out.

Line 488: Add reference to Gaydos and Strome, 2014 and to Kaneshiro et al 2022, which showed that H3K27me3 is transgenerationally inherited in *C. elegans*.

We have added these additional citations, now on line 557.

line 499: Does homology of most ortholog pairs extend well outside the set domain?

Yes, for most SET ortholog pairs, the homology does extend outside of the SET domain; this is particularly true within any other important binding or structural domains. However, it is not universal - there are a few ortholog pairs with orthology seemingly limited to the SET domain. Therefore, our observation of the lack of orthology outside the SET domain between the proteins clustering with MES-2 (line 567) is not iron-clad proof against orthology. But, together with our other evidence (low bootstrap value in phylogeny, synteny analysis of *mes-2*, etc.), it suggests that some similarity within the SET domains are facilitating a weak pairing of these two *P. pacificus* proteins with MES-2 (Figure S1a) – but they are not likely orthologs of MES-2. They are however obvious interesting candidates for follow-up investigation.

Figure 4E: There is a typo in the first column of Figure 4E, as it is labelled EZH2/MES-6.

We have made the appropriate correction and thank the reviewer for pointing it out.

Line 586: I didn't understand the point the authors were trying to make regarding the punctate nature of the H3K27me3 staining. What was meant by "punctate"? Did they mean that H3K27me3 did not cover all chromatin marked by H3? This is known to be the case in *C. elegans*.

Indeed, this is the point we were trying to make. To clarify we have changed the text on line 655 to read: "Furthermore, H3K27me3 does not appear to cover all chromatin marked by H3, suggesting it is enriched on specific chromosomes. Therefore, H3K27me3 is retained in the *P. pacificus* gonad, potentially for X chromosome silencing (as in *C. elegans*), despite the loss of PRC2."

Please add names at the top of all additional files to help the reader, and if possible, include a list of additional files and their descriptions somewhere in the paper

We have now added names and descriptions at the top of all files. We had previously listed descriptions of all additional files in the Data Availability section of the paper. However, to make these descriptions more conspicuous we have now added file descriptions throughout the Methods section.

Please provide legends for every table (e.g., on a separate tab), and spell out all abbreviations (e.g., HMT etc).

We have ensured that all tables in the revision have legends and have spelled out abbreviations (where applicable). We have also added descriptions and abbreviations to every Additional File in a separate tab.

March 5, 2024

RE: GENETICS-2024-306912

Dr. Michael S Werner
University of Utah
School of Biological Sciences
201 Presidents' Cir
Salt Lake City, Utah 84112

Dear Dr. Werner:

Congratulations! We are delighted to inform you that your manuscript entitled "Characterization of the *Pristionchus pacificus* "epigenetic toolkit" reveals the evolutionary loss of the histone methyltransferase complex PRC2" is acceptable for publication in GENETICS. Your response to the reviewers' comments, and the modifications to the manuscript were thorough and addressed all concern. Many thanks for submitting your intriguing research to the journal.

To Proceed to Production:

1. Format your article according to GENETICS style, as discussed at <https://academic.oup.com/genetics/pages/general-instructions>, and upload your final files at <https://genetics.msubmit.net>.
2. Your manuscript will be published as-is (unedited-as submitted, reviewed, and accepted) at the GENETICS website as an Advanced Access article and deposited into PubMed shortly after receipt of source files and the completed license to publish. Please notify sourcefiles@thegsajournals.org if you do not wish to publish your article via Advanced Access.
3. We invite you to submit an original color figure related to your paper for consideration as cover art. Please email your submission to the editorial office or upload it with your final files. You can submit a small-sized image for evaluation, and if selected, the final image must be a TIFF file 2513px wide by 3263px high (8.375 by 10.875 inches; resolution of 600ppi). Please avoid graphs and small type.

If you have any questions or encounter any problems while uploading your accepted manuscript files, please email the editorial office at sourcefiles@thegsajournals.org.

Sincerely,

Valerie Reinke
Associate Editor
GENETICS

Approved by:
Audrey Gasch
Senior Editor
GENETICS

note: Please add jnls.author.support@oup.com and genetics.oup@kwglobal.com (or the domains @oup.com and @kwglobal.com) to your email program's "safe senders" list. You will be contacted by both at various points during the production process.

Review comments (if applicable):